# Advanced glycation end-products reduce lipopolysaccharide uptake by macrophages

**Atsuhiro Kitaura**[1], **Takashi Nishinaka**[2], **Shinichi Hamasaki**[1], **Omer Faruk Hatipoglu**[2], **Hidenori Wake**[3], **Masahiro Nishibori**[3], **Shuji Mori**[4], **Shinichi Nakao**[1], **Hideo Takahashi**[2]*

**1** Department of Anesthesiology, Faculty of Medicine, Kindai University, Osaka-Sayama, Osaka, Japan, **2** Department of Pharmacology, Faculty of Medicine, Kindai University, Osaka-Sayama, Osaka, Japan, **3** Department of Pharmacology, Dentistry, and Pharmaceutical Sciences, Okayama University Graduate School of Medicine, Okayama, Japan, **4** Department of Pharmacy, Shujitsu University, Okayama, Japan

* hkt@med.kindai.ac.jp

**Data Availability Statement:** All relevant data are within the paper and its Supporting information files.

## Abstract

Hyperglycaemia provides a suitable environment for infections and the mechanisms of glucose toxicity include the formation of advanced glycation end-products (AGEs), which comprise non-enzymatically glycosylated proteins, lipids, and nucleic acid amino groups. Among AGE-associated phenotypes, glycolaldehyde-derived toxic AGE (AGE-3) is involved in the pathogenesis of diabetic complications. Internalisation of endotoxin by various cell types contributes to innate immune responses against bacterial infection. An endotoxin derived from Gram-negative bacteria, lipopolysaccharide (LPS), was reported to enhance its own uptake by RAW264.7 mouse macrophage-like cells, and an LPS binding protein, CD14, was involved in the LPS uptake. The LPS uptake induced the activation of RAW264.7 leading to the production of chemokine CXC motif ligand (CXCL) 10, which promotes T helper cell type 1 responses. Previously, we reported that AGE-3 was internalised into RAW264.7 cells through scavenger receptor-1 Class A. We hypothesized that AGEs uptake interrupt LPS uptake and impair innate immune response to LPS in RAW264.7 cells. In the present study, we found that AGE-3 attenuated CD14 expression, LPS uptake, and CXCL10 production, which was concentration-dependent, whereas LPS did not affect AGE uptake. AGEs were reported to stimulate the receptor for AGEs and Toll-like receptor 4, which cause inflammatory reactions. We found that inhibitors for RAGE, but not Toll-like receptor 4, restored the AGE-induced suppression of CD14 expression, LPS uptake, and CXCL10 production. These results indicate that the receptor for the AGE-initiated pathway partially impairs the immune response in diabetes patients.

## Introduction

Lipopolysaccharide (LPS), a potent macrophage activator, activates two signalling pathways, the Toll-like receptor 4 (TLR4)/myeloid differentiation primary response 88 (MyD88)/nuclear factor kappa B (NF-κB) pathway and the TLR4/TIR domain containing adaptor inducing

**Funding:** This work was supported by the Japan Society for the Promotion of Science (JSPS) Grants-in-Aid for Early-Career Scientists (17K16766), Scientific Research (C) (18K06905), by Japan Agency for Medical Research and Development (AMED) (15LK0201014h0003), by the Ministry of Education, Culture, Sports, Science and Technology (MEXT)-Supported Program for the Strategic Research Foundation at Private Universities (S1411037). The funders had no role in study design, data collection and analysis, decision to publish, or preparation of the manuscript.

**Competing interests:** The authors declare no competing interests.

interferon β (TRIF)/IFN regulatory factor (IRF3) pathway, on the plasma membrane and endosomal membrane, respectively [1]. When LPS binds to CD14 on the plasma membrane it is transferred to myeloid differentiation factor 2 (MD-2), which induces the dimerisation of TLR4/MD-2 [2]. Subsequently, the dimerised TLR4/MD-2 complex binds to LPS, which activates MyD88-dependent signalling and induces the production of proinflammatory cytokines, such as interleukin-1β (IL-1β), IL-6, and tumour necrosis factor-α, via the activation of NF-κB. Macrophages/monocytes were reported to recognise and take up LPS into intracellular vesicles, suggesting LPS detoxification [3]. The dimerised TLR4/MD-2 complex that binds to LPS is internalised and forms an endosome mediated by CD14. Ultimately, this complex on the endosomal membrane activates TRIF-dependent signalling and facilitates the transcription of interferon (IFN) and IFN-inducible genes, such as CXC chemokine ligand 10 (CXCL10/ IFN inducible protein 10, IP10), mediated by the phosphorylation of IRF3. Taken together, these results indicate that CD14 plays important roles in the LPS-initiated signalling pathway for inflammatory responses of macrophages.

Prolonged exposure to hyperglycaemia is the primary cause of most diabetic complications such as impaired immune responses to microbes [4]. Infectious disease and subsequent sepsis, which cause multiple organ dysfunction by inappropriate immune responses to invading microbes, account for most deaths of patients with diabetes [5, 6]. Hyporesponsiveness to LPS is characterised as endotoxin tolerance, where innate immune cells exposed to low concentrations of endotoxin are unable to respond to further challenges with endotoxin, which is also observed in sepsis [7]. Patients with type 2 diabetes showed decreased cytokine release responses to LPS and LPS clearance [8–10].

The formation of endogenous advanced glycation end-products (AGEs) occurs non-enzymatically between free carbonyls of sugars and amino groups of macromolecules by the Maillard reaction. In turn, the accumulation of AGEs in long-lived tissue proteins of patients with diabetes induces inflammatory mechanisms in tissues. Serum AGEs levels were reported to be elevated over 2-fold in patients with diabetes (approximately 25 μg/ml) and were increased almost 8-fold in patients with diabetes requiring haemodialysis (approximately 80 μg/ml) when compared with healthy individuals [11, 12]. In particular, a toxic AGE structure termed glycolaldehyde-derived AGE (AGE-3) is involved in the pathogenesis of renal failure, arteriosclerosis, angiopathy, and retinopathy in these patients [13]. AGEs interact with two types of cell surface AGE receptors expressed on macrophages [14]. A scavenger receptor, scavenger receptor-1 class A (SR-A)/CD204, was involved in AGEs uptake by a mouse macrophage cell line, RAW264.7 cells, despite its physiological role remain unclear [15]. Among constitute other types of AGE receptors, receptor for AGE (RAGE) and TLR4 initiate specific cellular signalling events.

Cross tolerance, which priming of macrophage with AGEs show decreased response to LPS, has been established [16], indicating that AGEs is involved in impaired immune responses to microbes. However, the mechanisms by which AGEs induce impaired immune responses in macrophages is poorly understood. We hypothesized that AGEs uptake interrupt LPS uptake and impair innate immune response to LPS in macrophage. In the present study, we found that toxic AGE inhibited LPS uptake and CXCL10 release via the suppression of CD14 expression in RAW264.7 cells. RAGE was involved in the AGE-3-induced suppression of CD14 expression, LPS uptake, and CXCL10 release. Because CXCL10 promotes Th1 responses [38], the suppression of CXCL10 production might be a mechanism to evade host immunity. Thus, these findings suggest that the blockade of AGE-RAGE signalling normalises innate immune responses by macrophages in a diabetes environment.

## Materials and methods

### Reagents and antibodies

LPS from *Escherichia coli* O111:B4 was purchased from Sigma-Aldrich (St Louis, MO, USA). Alexa Fluor 488- and 594-conjugated LPS from *Escherichia coli* were purchased from Invitrogen (488: L23351, 594: L23353, Carlsbad, CA, USA). AGE-modified bovine serum albumin (BSA) was prepared as previously described [15]. Briefly, 50 mg/mL of BSA (Fujifilm Wako, Osaka, Japan) was incubated under sterile conditions with 0.2 M of glycolaldehyde (AGE-3) (Sigma-Aldrich) in 0.2 M of phosphate buffered saline (PBS) (pH 7.4) at 37°C for 7 days. BSA incubated under the same conditions was used as a control. Excess glycolaldehyde removed from AGE-BSA or BSA solution by dialysis for 2 days at 4°C. Cellulose tube 27/32 (UC27-32-100, SEKISUI CHEMICAL, Osaka, Japan) was filled with AGE-BSA or BSA solution and dialysed in 0.02 M of PBS. The endotoxin concentration was 1.2 pg/ml in 100 μg/ml of AGE (measured at SRL, Okayama, Japan). The following pharmacological inhibitors and neutralising antibodies were used: sucrose (0.45 M, Fujifilm Wako), chlorpromazine (20 μM, C2481, Tokyo Chemical Industry, Tokyo, Japan), genistein (40 μM, G0272, Tokyo Chemical Industry), FPS-ZM1 (0.25–1 μM, 553030, Millipore, Burlington, MA, USA), LPS-RS (10 μg/mL, tlrl-prslps, InvivoGen, San Diego, CA, USA), and neutralising antibodies against RAGE (polyclonal Goat IgG, final concentration; 20 μg/ml, AF1179), LOX-1 (polyclonal Goat IgG, final concentration; 20 μg/ml, AF1564), SR-A/CD204 (polyclonal Goat IgG, final concentration; 20 μg/ml, AF1797, all R&D Systems, Minneapolis, MN, USA), CD14 (monoclonal Rat IgG$_{2b}$ κ, clone; 4C1/CD14, final concentration; 20 μg/ml, 557896, BD Biosciences, Franklin Lakes, NJ, USA), CD36 (monoclonal Mouse IgG$_{2a}$ κ, clone; 185-1G2, final concentration; 20 μg/ml, MA5-14112, Thermo Fisher Scientific, Waltham, MA, USA), TLR4 (monoclonal Rat IgG$_{2a}$ κ, clone; MTS510, final concentration; 20 μg/ml, 117608, BioLegend, San Diego, CA, USA), TLR5 (monoclonal Mouse IgG$_{2a}$ κ, clone; 19D759.2, final concentration; 20 μg/ml, NBP2-24787, Novus Biologicals, Centennial, CO, USA) and TLR7 (monoclonal Mouse IgG$_1$ κ, clone; 4G6, final concentration; 20 μg/ml, NBP2-27332, Novus Biologicals). Anti-mouse Abs against phycoerythrin (PE)-conjugated SR-A (monoclonal Human IgG$_1$, clone; REA148, final concentration; 4 ng/test, 130-102-328, Miltenyi Biotec, Bergisch Gladbach, Germany), PE-conjugated TLR4 (monoclonal Mouse IgG$_1$ κ, clone; UT41, final concentration; 50 ng/test, 12-9041-80, Thermo Fisher Scientific), PE-conjugated LOX-1 (monoclonal Mouse IgG$_{2a}$, clone; 214012, final concentration; 0.5 μl/test, FAB1564P, R&D Systems), allophycocyanin (APC)-conjugated CD14 (monoclonal Rat IgG$_{2a}$ κ, clone; Sa14-2, final concentration; 50 ng/test, 123312, BioLegend), APC-conjugated CD36 (monoclonal Armenian Hamster IgG, clone; HM36, final concentration; 25 ng/test, 102612, BioLegend), or APC-conjugated RAGE (polyclonal Rabbit IgG, final concentration; 50 ng/test, LS-C212626, LifeSpan BioSciences, Seattle, WA, USA) were used for FACS analysis. Isotype negative control antibodies against anti-CD14 antibody (Rat IgG$_{2b}$ κ, final concentration; 20 μg/ml, 553986, BD Biosciences, Franklin Lakes, NJ, USA) and anti-RAGE antibody (Goat IgG, final concentration; 20 μg/ml, sc-2028, Santa Cruz Biotechnology, Dallas, TX, USA) were used.

### Cell culture and stimulation

The mouse macrophage cell line RAW264.7 (EC91062702, DS Pharma Biomedical, Osaka, Japan) was grown in Dulbecco's modified Eagle medium containing 2 mM glutamine and 10% heat-inactivated foetal bovine serum at 37°C and 5% $CO_2$. RAW264.7 cells were plated in 24-well plates at 5–10$^5$ cells/well. After RAW264.7 cells had adhered to the plate, they were

incubated with different doses of AGEs (2, 20, and 200 μg/ml) and LPS (1, 10, 100 ng/mL, and 1 μg/mL) for the indicated times according to each experiment.

## Fluorescent labelling of BSA and AGE-3 using Alexa Fluor 488 C5 maleimide

AGE-3 was fluorescently labelled as described previously [15, 17]. Briefly, each protein was incubated with 20 times the amount of Alexa Fluor 488 C5 maleimide (A10254, Thermo Fisher Scientific) at room temperature for 2 h in PBS. Excess labelling reagent is removed by dialysed with PBS at 4°C for 2 days. Total protein concentration was quantified by the Bradford method using a Bradford protein assay kit (Bio-Rad Laboratories, Kidlington, UK). Alexa Fluor 488-labelled compound fluorescence intensity was measured using ARVO MX 1420 (PerkinElmer Japan, Yokohama, Japan) (excitation: 485 nm, emission: 535 nm). The strengths of Alexa Fluor 488-BSA or -AGE-3 per unit dosage were adjusted by adding respective unlabelled proteins.

## Flow cytometric analysis of the cellular uptake of LPS and AGE-3

Flow cytometric analysis of the cellular uptake of LPS and AGE-3 was performed as described previously with some modifications [15]. Briefly, RAW264.7 cells were seeded at $2 \times 10^4$ cells/well in 24-well plates. After overnight incubation, cells were treated with Alexa Fluor 488-labelled AGE-3 at concentrations of 2 to 200 μg/ml and/or Alexa Fluor 594-labelled LPS at concentrations of 1 ng/ml to 1 μg/mL for 1, 2, or 4 h. For experiments using neutralising antibodies, cells were pre-treated with neutralising antibody against each receptor for 1 h before incubation with LPS and/or AGE-3. For experiments using pharmacological inhibitors, cells were concomitantly treated with each pharmacological inhibitor and LPS and/or AGE-3. Subsequently, cells were mechanically detached by pipetting and processed twice by rinsing with FACS wash buffer consisting of PBS supplemented with 2.5% normal horse serum, 0.1% sodium azide, and 10 mM HEPES followed by centrifugation ($200 \times g$, 5 min, 4°C). Subsequently, 300 μl PBS (−) was added to the residue and cells were stained with DRAQ7™ (3 μM, Biostatus, Shepshed, UK) to exclude dead cells. Thereafter, the analysis was performed using an LSRFortessa™ X-20 cell analyser (BD Biosciences) or FACS Aria (BD Biosciences) and the data were processed using BD FACSDiva software (BD Biosciences) to determine the mean fluorescence intensity (MFI) of Alexa Fluor 488-labelled AGEs and Alexa Fluor 594-conjugated LPS.

## Confocal fluorescence microscopy for macrophage uptake of LPS

RAW264.7 cells seeded at $4.0 \times 10^5$ cells/dish in 35-mm glass-bottom dishes (Matsunami Glass, Kishiwada, Japan) were allowed to attach for 1 h followed by treatment with Alexa Fluor 488-labelled LPS and AGE-3 at 200 μg/ml for 4 h. The fluorescence images of Alexa Fluor 488-labelled LPS were captured at an original magnification of ×400 using a confocal laser C2 microscope (Nikon, Tokyo, Japan).

## Flow cytometric analysis of the surface receptors on RAW264.7 cells

Flow cytometric analysis of the expression of cell surface receptors was performed as described above with some modifications. Briefly, RAW264.7 cells were seeded at $2 \times 10^4$ cells/well in 24-well plates. After overnight incubation, cells were treated with LPS at 1 μg/mL for 1, 2, or 4 h. Subsequently, cells were mechanically detached by pipetting and rinsed with FACS wash buffer followed by centrifugation ($200 \times g$, 5 min, 4°C), then stained with each antibody at 4°C

for 30 min. After rinsing with FACS wash buffer followed by centrifugation ($200 \times g$, 5 min, 4˚C), 300 μl of PBS (−) was added to the residue followed by staining with PI (2 μg/ml). The MFI of each surface membrane receptor was measured using an LSRFortessa™ X-20 cell analyser and BD FACSDiva software for data processing.

## Quantitation of CXCL10 in cell culture supernatants

CXCL10 released from RAW264.7 cells into culture supernatants was measured by an enzyme-linked immunoassay (DY466, R&D Systems). RAW264.7 cells were seeded at $5.0 \times 10^4$ cells/well in 24-well plates and incubated overnight. Then, cells were treated with LPS at 1 μg/mL for 24 h. In some experiments, neutralising Abs or a pharmacological inhibitor were added concomitantly with LPS. After incubation, cell culture supernatants were obtained by centrifuging medium at $200 \times g$, 5 min, 4˚C. Quantitation of CXCL10 by enzyme-linked immunoassay was performed according to the manufacturer's instructions.

## Cell viability

RAW264.7 cells were seeded at $2.0 \times 10^4$ cells/well in 96-well plates, then treated with LPS at 1 μg/mL and/or AGE-3 at 200 μg/mL. Cell viability was analysed using a Cell Counting Kit-8 (Dojindo Laboratories) according to the manufacturer's instructions. The absorbance was measured using a microplate reader (Model680, Bio-Rad).

## Preparation for the whole cellular lysates of RAW264.7 cells and mouse lung tissue

RAW264.7 cells were seeded at $2.0 \times 10^5$ cells/well in 6-well plates and incubated for 24 h at 37˚C under 5% $CO_2$. Then, cells were washed with PBS, mechanically detached by pipetting. Mouse lung tissue were obtained from male ICR mice (SLC, Hamamatsu, Japan). Lung tissue were disrupted by the ultrasonic disrupter (TOMY, Tokyo, Japan). The cells and lung tissue were lysed in radio-immunoprecipitation assay buffer containing protease inhibitors (Antipain; #4062 and Leupeptin; #4041, PEPTIDE INSTITUTE INC., Osaka, Japan, Aprotinin; #033–46, nacalai tesque, Kyoto, Japan, concentration of each reagents; 2.0 μg/mL) and phosphatase inhibitor (4906837001, Sigma-Aldrich) for 30 min on ice. The supernatant of the resulting suspension was obtained after centrifugation ($16,000 \times g$, 20 min, 4˚C) and collected as the total cell lysate. The total protein concentration was quantified using a Bradford protein assay kit. Animal experiments were approved by the institutional guidelines of the Centre for Experimental Animal Care and Use, Kindai University Faculty of Medicine (KAME-26-040).

## Western blot analysis

The cell or tissue lysate were mixed with an equal volume of 2 × sample buffer containing 0.5 M Tris-HCl (pH 6.8), 15% sodium dodecyl sulphate (SDS), 12% β-mercaptoethanol, 20% glycerol, and 0.1% bromophenol blue, then heated for 5 min at 97˚C. Protein samples (10 μg/lane) was subjected to SDS-polyacrylamide gel electrophoresis followed by electro transfer onto a nitrocellulose membrane. The blotted membranes were blocked at room temperature for 1 h with blocking buffer containing 5% non-fat dry milk (198–10605, Fujifilm Wako, Osaka, Japan) for GAPDH, or 5% BSA (015–15103, Fujifilm Wako) for RAGE in Tris-buffered saline (TBS)-T (TBS pH 7.6 with 0.1% Tween-20). The membrane was then probed with primary rabbit anti-RAGE (ab3611, 1:2,000 dilution, abcam, Cambridge, UA) or GAPDH (MAB374, 1:20,000 dilution, Merck Millipore, Darmstadt, Germany) used as internal control in respective blocking buffer at 4˚C overnight. Blots were then washed with TBS-T and incubated with

horseradish peroxidase-conjugated anti-rabbit secondary Abs (074–1506, 1:4,000 dilution, Kirkegaad and Perry Laboratories, Guildford, UK) or anti-mouse secondary Abs (074–1806, 1:4,000 dilution, Kirkegaad and Perry Laboratories) in respective blocking buffer for 1 h at room temperature. After rinsing with TBS-T, the immune-complexes were visualized using PierceTM ECL Western Blotting Substrate (Thermo Fisher Scientific). Immunoreactive bands were visualized using an AmershamTM Imager 600 CCD-based chemiluminescent analyser (GE Health Care Japan, Tokyo, Japan).

## Statistical analyses

Data are presented as the means ± SEM. Sample size means the number of independent experiments or replicates. Statistical analyses were performed using Prism version 7 software (GraphPad Software, La Jolla, CA, USA). Statistical significance was assessed using a one-way analysis of variance followed by Student's *t*-test for comparisons between two groups or Dunnett's or Tukey's post hoc test for comparisons between more than three groups. A *p*-value < 0.05 was considered statistically significant.

## Results

### Effect of LPS and AGE-3 on the viability of RAW264.7 cells

Immune cell defects are thought to contribute to the increased susceptibility of patients with diabetes to bacterial infections [4, 18]. In the present study, we investigated the effect of AGEs on the responses of RAW264.7 cells to LPS. We confirmed that the single or combined administration of LPS and AGE-3 at 1 μg/ml and 200 μg/ml respectively, the maximum concentrations used in the present study, did not affect the viability of RAW264.7 cells (S1 Fig).

### Concentration- and time-dependent uptake of LPS by RAW264.7 cells

In a previous study, we found that AGEs were concentration- and time-dependently internalised into RAW264.7 cells [15]. The direct fluorometric measurement of AGEs in body fluids used excitation wavelengths in the range of 350–390 nm [19]. To reduce the effect of changes in fluorometric intensity by AGE uptake, we used Alexa Fluor 594-labelled LPS to assess the LPS uptake in RAW264.7 cells by means of quantitative flow cytometry in experiments other than Fig 1D. In Fig 1D, we used Alexa Fluor 488-labelled LPS to assess the LPS uptake.

A representative histogram plot shows the uptake of LPS at a concentration of 1 μg/ml after 4 h treatment (Fig 1A). Fig 1B shows concentration-dependent changes in the uptake of fluorescently-labelled LPS at concentrations ranging from 1 ng/ml to 1 μg/ml by RAW264.7 cells at 4 h after treatment. We found that LPS at 100 ng/ml or 1 μg/ml induced its internalisation by RAW264.7 cells, whereas doses of 1 and 10 ng/ml were not internalised. Next, we examined time-dependent changes in the uptake of fluorescently-labelled LPS at 1 μg/ml by RAW264.7 macrophage cells at 1 h to 4 h after treatment (Fig 1C). The mean florescence intensity (MFI) indicating uptake of fluorescently-labelled LPS at 1 μg/ml increased at 2 h by 268% and 4 h by 350%. As shown in Fig 1D, the endocytic uptake of fluorescently-labelled LPS by RAW264.7 cells were confirmed by confocal fluorescence microscopy. Of multiple different endocytic pathways, clathrin-mediated endocytosis is the major pathway for the concentrative uptake of receptors and receptor–ligand complexes [20, 21]. Clathrin-independent endocytosis includes the constitutive pinocytotic pathway as well as endocytosis mediated by caveolae and glyco-lipid rafts [22, 23]. In the present study, we found that LPS uptake by RAW264.7 cells were decreased by clathrin-mediated endocytosis inhibitors, sucrose at 0.45 M by 69.6% and chlor-promazine at 20 μM by 57.8% (Fig 1E). However, an inhibitor of caveola-mediated endocytosis

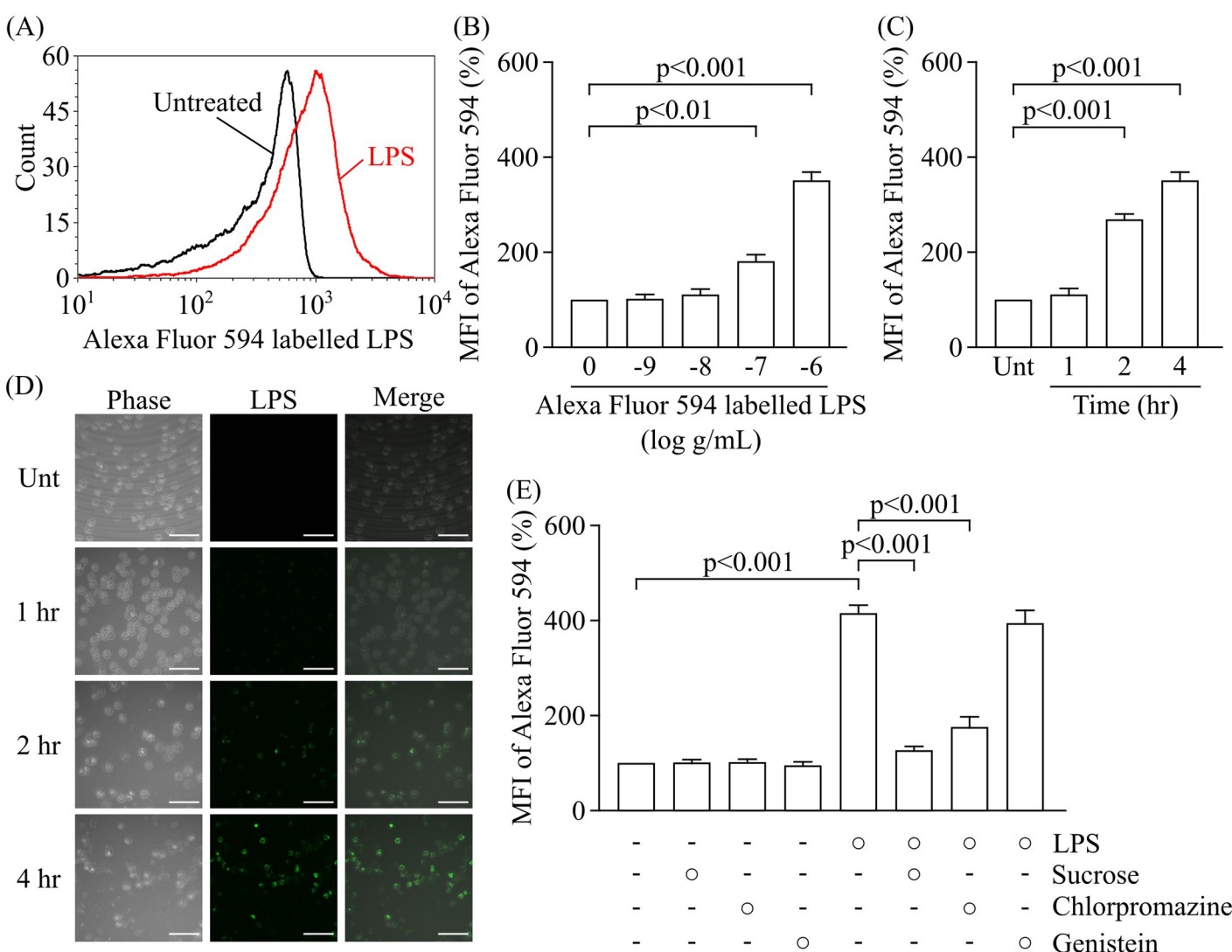

**Fig 1. LPS uptake is mediated by endocytosis in RAW264.7 cells.** Cells were treated with fluorescence-labelled LPS and analysed by flow cytometry. (A) Representative histogram of fluorescence intensity for Alexa Fluor 594-labelled LPS in RAW264.7 cells treated with medium alone (Untreated; greyscale) and Alexa Fluor 594-labelled LPS at 1 μg/mL for 4 h (red line). (B) Changes in mean fluorescence intensity (MFI) of Alexa Fluor 594 in RAW264.7 cells treated with Alexa Fluor 594-labelled LPS at concentrations ranging from 1 ng/mL to 1 μg/mL for 4 h (n = 3 means the number of independent experiments, Dunnett's test). (C) Time course of MFI in RAW264.7 cells treated with Alexa Fluor 594-labelled LPS at 1 μg/mL (n = 3, Dunnett's test). (D) Representative images of LPS uptake visualised by Alexa Fluor 488-labelled LPS (green) in RAW264.7 cells. Scale bars: 50 μm. (E) MFI of Alexa Fluor 594-labelled LPS at 1 μg/mL in RAW264.7 cells treated with endocytosis inhibitors for 1 h (n = 3, Tukey's test). Each column represents the MFIs of Alexa Fluor 594-labelled LPS relative to the concentration at "0" or medium alone (Unt), which was arbitrarily defined as 100%. Data are presented as the means ± SEM. MFI; mean fluorescence intensity. Unt; untreated.

and lipid raft-mediated endocytosis, genistein (40 μM), had no effect on LPS uptake, indicating that LPS-uptake was mediated by clathrin-mediated endocytosis.

## The involvement of CD14 in LPS uptake by RAW264.7 cells

CD14 is an essential protein for LPS recognition and uptake. In contrast to CD14, TLR4 contributes to the activation of signalling responses to LPS, although it is not required for LPS uptake [24].

We confirmed changes in the expression of CD14 on the plasma membrane after exposure to LPS. CD14 expression was significantly decreased at 1 h and 2 h after treatment with LPS at

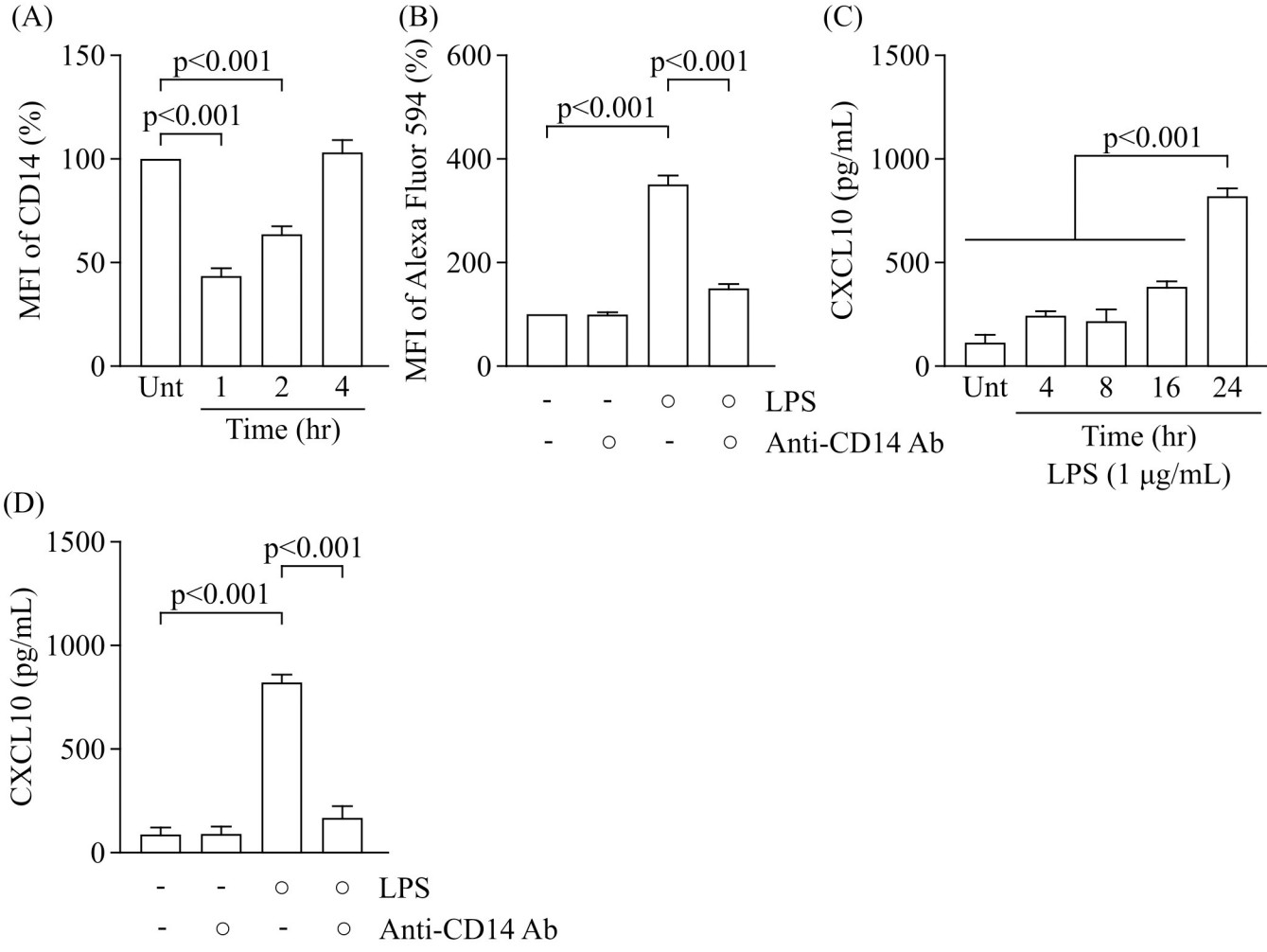

**Fig 2. Involvement of CD14 on LPS uptake and CXCL10 release in response to LPS.** (A) Time course of MFI related to CD14 in RAW264.7 cells treated with LPS at 1 μg/mL (n = 3, Dunnett's test). (B) Effect of anti-CD14 antibody at 20 μg/mL on LPS uptake (n = 3, Tukey's test). Neutralising antibody was added 1 h before treatment with Alexa Fluor 594-labelled LPS at 1 μg/mL and incubated for 4 h. MFI relative to the untreated group (medium alone) was arbitrarily defined as 100%. (C) Time course of CXCL10 levels in the culture supernatants obtained from cells treated with LPS at 1 μg/mL (n = 6, Dunnett's test). (D) Effect of anti-CD14 antibody at 20 μg/mL on CXCL10 release from RAW264.7 cells treated with LPS at 1 μg/mL (n = 3, Tukey's test). CXCL10 levels in the culture supernatants obtained from cells treated with LPS and/or anti-CD14 antibody for 24 h were measured by ELISA. Data are presented as the means ± SEM. MFI; mean fluorescence intensity. Unt; untreated. Ab; antibody.

1 μg/ml (Fig 2A). This was restored to basal levels at 4 h after LPS treatment indicating that exposure to LPS caused the internalisation of CD14. Previous studies reported that neutralising antibody against CD14 (clone 4C1) inhibited the binding of LPS to CD14 and suppressed LPS-induced cytokine release [25, 26]. Consistent with previous reports, we confirmed that pretreatment with anti-CD14 antibody at 20 μg/ml reduced the LPS uptake by 57.1% (Fig 2B).

LPS uptake mediated by CD14 leads to the activation of TRIF signalling and upregulation of type 1 IFN-inducible cytokines including IFN-inducible protein 10 (IP-10/CXCL10). We confirmed that the remarkably increased CXCL10 production in the culture supernatants were able to be detected by exposure to LPS for 24 h but not 4, 8 and 16 h (Fig 2C). Previous report have demonstrated that upregulation of CXCL10 mRNA by LPS began to increase and reached maximal level at 4 h in peritoneal macrophage prepared from mouse [27]. And then CXCL10 protein are produced by translation and released into culture medium, indicating

that the time lag is occurred between CD14-initiated LPS uptake and CXCL10 production. In experimental condition which is exposed to LPS for 24 h, we found that anti-CD14 antibody at 20 μg/ml also suppressed the increased release of CXCL10 in LPS-treated RAW264.7 cells by 79.5% (Fig 2D). Scavenger receptors are also involved in LPS uptake by macrophages [28]. Although LPS increased the levels of scavenger receptors including lectin-like oxidised low-density lipoprotein receptor 1 (LOX-1) and receptor for AGE (RAGE), but not SR-A (CD204) and CD36 (S2 Fig), each neutralising antibody failed to inhibit LPS uptake (S3 Fig). In addition, neutralising antibodies against TLR4, TLR5, and TLR7 had no effect on LPS uptake (S3 Fig), whereas TLR4 levels were upregulated by LPS (S2 Fig). An isotype-matched control for anti-CD14 Ab had no effect on LPS uptake in each treatment group (S4 Fig). Thus, CD14 is a critical mediator of LPS uptake and LPS-induced release of CXCL10 from RAW264.7 cells.

## The effect of AGE-3 on LPS uptake by RAW264.7 cells

It was reported that initial AGE-modified BSA priming mediated tolerance against LPS [16]. The effect of AGE-3 at concentrations ranging from 0.1 to 200 μg/ml was determined by *in vitro* studies using macrophages [29, 30]. In a previous study, we also demonstrated that toxic AGE (AGE-3) at 20 and 200 μg/ml enhanced its own uptake by RAW264.7 cells [15].

We analysed the uptake of fluorescence-labelled LPS in the absence or presence of AGE-3 at concentrations ranging from 2 to 200 μg/ml by flow cytometry (Fig 3A). As shown in Fig 3B, treatment with AGE-3 at 20 and 200 μg/ml for 4 h significantly inhibited LPS uptake by RAW264.7 cells, and the LPS uptake was decreased by almost 25% in the presence of AGE-3 at 200 μg/ml. AGE-3 at 200 μg/ml inhibited LPS uptake in RAW264.7 cells as observed by confocal fluorescence microscopy (Fig 3C). Although AGEs competitively antagonised LPS uptake, LPS and anti-CD14 antibody had no effect on AGE uptake in RAW264.7 cells (S5 Fig).

Next, we examined the concentration- and time-dependent changes in the expression of CD14 from 1 h to 4 h after treatment in the presence of AGE-3 at concentrations ranging from 2 to 200 μg/ml. AGE-3 at 200 μg/ml inhibited the expression of CD14 on RAW264.7 cells at 4 h (Fig 3D and 3E). In the presence of LPS, AGE-3 at 200 μg/ml also suppressed the expression of CD14 at 1, 2 and 4 h after treatment (Fig 3F). Moreover, AGE-3 at 200 μg/ml also reduced CXCL10 production in the presence of LPS at 1 μg/ml by 62% (Fig 3G). These results suggest that the AGE-3-induced downregulation of CD14 expression partially contributed to the reduction of LPS uptake and CXCL10 production in response to LPS.

## RAGE partly mediates the effect of AGE on LPS uptake by RAW264.7 cells

AGEs mainly bind to RAGE to facilitate inflammatory NF-κB signalling in macrophages [31], indicating that the AGE/RAGE axis drives proinflammatory signalling. AGEs also able to interact with TLR4 [32]. In addition, we previously reported that, among scavenger receptors, AGE-3 increased the expression of SR-A on RAW264.7 cells and that the AGE uptake was inhibited by a neutralising antibody against SR-A [15], indicating the involvement of SR-A in AGE uptake and the AGE-induced activation of monocytes.

In this study, we examined which receptor contributed to the inhibitory effect of AGE-3 on the response to LPS in RAW264.7 cells. A RAGE antagonist, FPS-ZM1 at 0.5 and 1 μM, and neutralising antibody against RAGE at 20 μg/mL prevented the AGE-3-induced downregulation of CD14 expression (Fig 4A and 4B). There was no significant effect of LPS-RS, a TLR4 antagonist, at 10 μg/mL or a neutralising antibody against SR-A (CD204) at 20 μg/mL on CD14 expression and LPS uptake in AGE-3-treated RAW264.7 cells. However, anti-SR-A antibody suppressed the uptake of AGE-3 in RAW264.7 cells (S5 Fig) indicating that the concentration of anti-SR-A antibody sufficient to suppress effects mediated by AGE-3. Therefore,

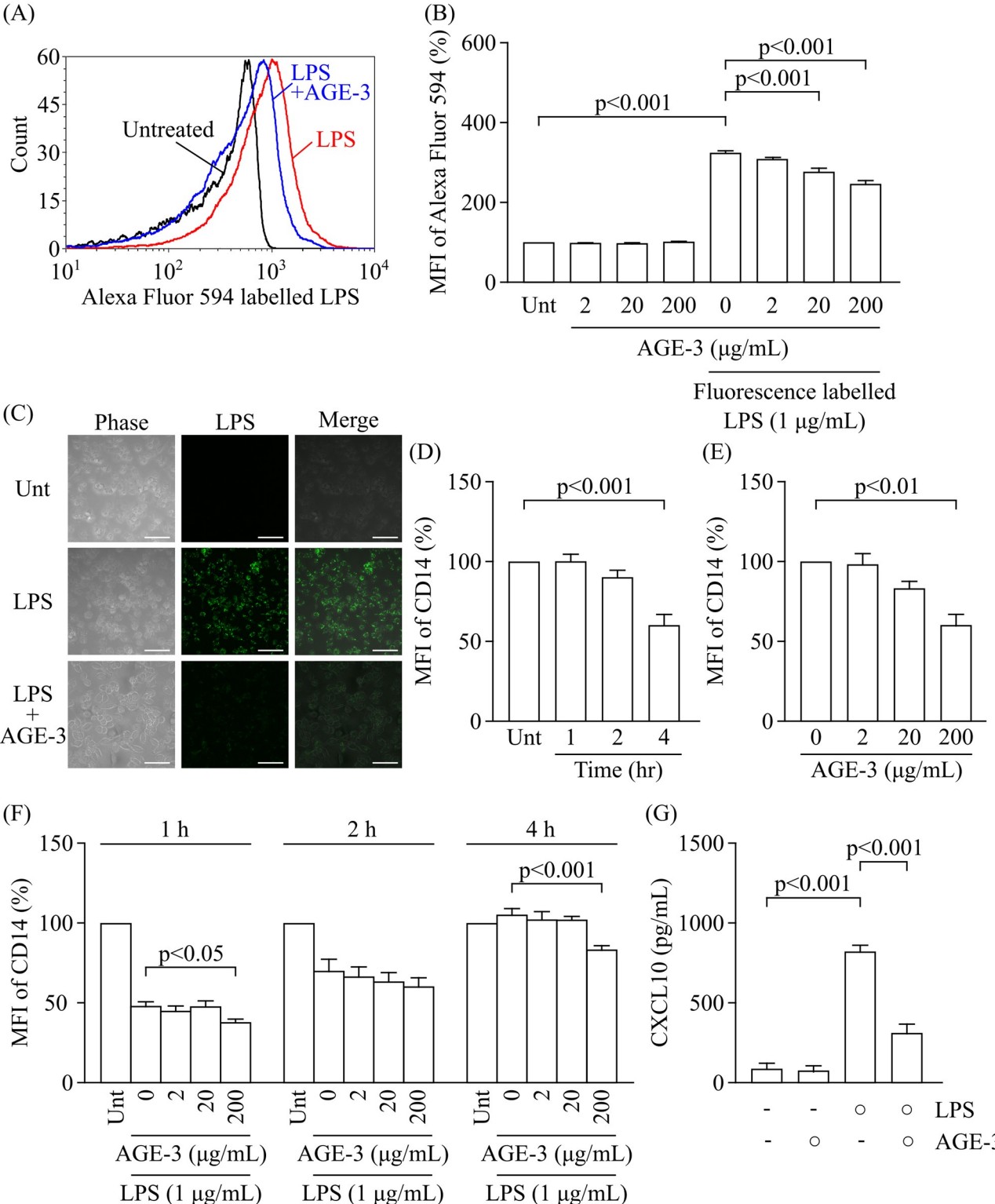

**Fig 3. Inhibitory effect of AGE-3 on the cellular response to LPS in RAW264.7 cells.** (A) Representative histogram of fluorescence intensity for Alexa Fluor 594-labelled LPS in RAW264.7 cells treated with medium alone (Untreated; greyscale), Alexa Fluor 594-labelled LPS at 1 μg/mL (red line), and a combination of Alexa Fluor 594-labelled LPS and AGE-3 at 200 μg/mL (blue line). (B) Effect of AGE-3 on LPS uptake (n = 3, Tukey's test). Cells were concomitantly treated with Alexa Fluor 594-labelled LPS at 1 μg/mL and AGE-3 at 200 μg/mL for 4 h. MFI relative to the untreated group (medium alone) was arbitrarily defined as 100%. (C) Representative images of LPS uptake visualised by Alexa Fluor 488-labelled LPS (green) in

RAW264.7 cells treated with AGE-3 at 200 μg/mL for 4 h. Scale bars: 50 μm. (D) Time course of MFI related to CD14 in RAW264.7 cells treated with AGE-3 at 200 μg/mL (n = 3, Dunnett's test). (E) Changes in MFI related to CD14 in RAW264.7 cells treated with AGE-3 at concentrations ranging from 2 to 200 μg/mL for 4 h (n = 3, Dunnett's test). (F) Effect of AGE-3 on the time course of changes in CD14 expression in RAW264.7 cells treated with LPS at 1 μg/mL (n = 5, Dunnett's test). (G) Effect of AGE-3 at 200 μg/mL on CXCL10 release from RAW264.7 cells treated with LPS at 1 μg/mL (n = 5, Tukey's test). CXCL10 levels in the culture supernatants obtained from cells treated with LPS and/or AGE-3 for 24 h were measured by ELISA. Data are presented as the means ± SEM. Unt; untreated.

SR-A do not contribute to the AGE-3-induced suppression of LPS uptake and CD14 expression. Consistent with this, the AGE-3-induced reduction of LPS uptake and CXCL10 release were significantly inhibited by treatment with FPS-ZM1 and anti-RAGE neutralising antibody (Fig 4C–4F). We verified the protein expression of RAGE in RAW264.7 cells by using western blot. Consistent with previous study [33], the two bands, which represent pre and post glycation type of RAGE protein, in the vicinity of 50 kDa were observed in mouse lung tissue used as positive control (S6 Fig). In contrast, the protein sample prepared from RAW264.7 cells showed weak band in the vicinity of 50 kDa. Taken together, these findings suggest that AGE-3 attenuated the response to LPS in RAW264.7 cells, which was partially mediated by the activation of RAGE (Fig 5).

## Discussion

On the basis of accumulating evidence that chronic inflammation, characterised as systemic low-grade inflammation induced by the activation of innate immune cells, provides a basis for understanding the pathophysiology of diabetes mellitus [34] and induces immunosuppression after infection [35, 36], we examined the regulation of macrophage activation. In the present study, we used AGE-3 at 200 μg/mL to examine the inhibitory effect of AGE-3 on the reaction response to LPS. We have no information about the AGE-3 levels in patients with diabetes. Measurement of AGE-3 have not been established. Serum AGEs levels are elevated 2~8-fold (almost 20~80 μg/mL) in patients with diabetes compared with healthy individuals [11, 12]. Therefore, it seems that AGE-3 levels are lower than 80 μg/mL. AGEs are gradually accumulated in tissue, raising the possibility that macrophage within the local microenvironment are exposed to high AGEs levels in patients with diabetes. Previously, we reported that a toxic AGE, AGE-3, at 200 or 500 μg/mL enhanced the activation of RAW264.7 macrophage-like cells after 4 or 1 h treatment by stimulating SR-A or TLR2/4, respectively, but not RAGE [15, 37]. In the present study, we found that AGE-3 at 200 μg/mL decreased the activation of RAW264.7 cells in the presence of LPS at 1 μg/mL after 4 h of treatment by stimulating RAGE, but not SR-A or TLR2/4. Although we have not clearly detected RAGE protein in RAW264.7 cells compared to lung tissue as positive control, AGE-3-induced the suppression of response to LPS was attenuated by FPS-ZM1 and neutralising antibody against RAGE. The RAGE expression in RAW264.7 cells may therefore be quite low. However, the low protein expression of RAGE did not result in a lower contribution of this receptor to AGE stimulation. Moreover, the duration of AGE exposure to each receptor was similar. Therefore, differences in the results obtained in the presence or absence of LPS might be associated with the deformations of the structure of AGE, suggesting it will be difficult to evaluate the affinity of AGE to each receptor. Furthermore, different effect of AGEs in the presence or absence of LPS may contribute to the acute inhibitory effect of AGEs on LPS uptake and CD14 expression. Therefore, it is very important to investigate differences in the interaction of AGEs to their putative receptors when determining which receptors are responsible for AGE recognition. More detailed investigations are needed.

The enhanced uptake of LPS attenuated inflammation and tissue damage [38, 39]. Although AGE-3-induced inhibition of LPS uptake was little (24% inhibition), the suppressive effect of

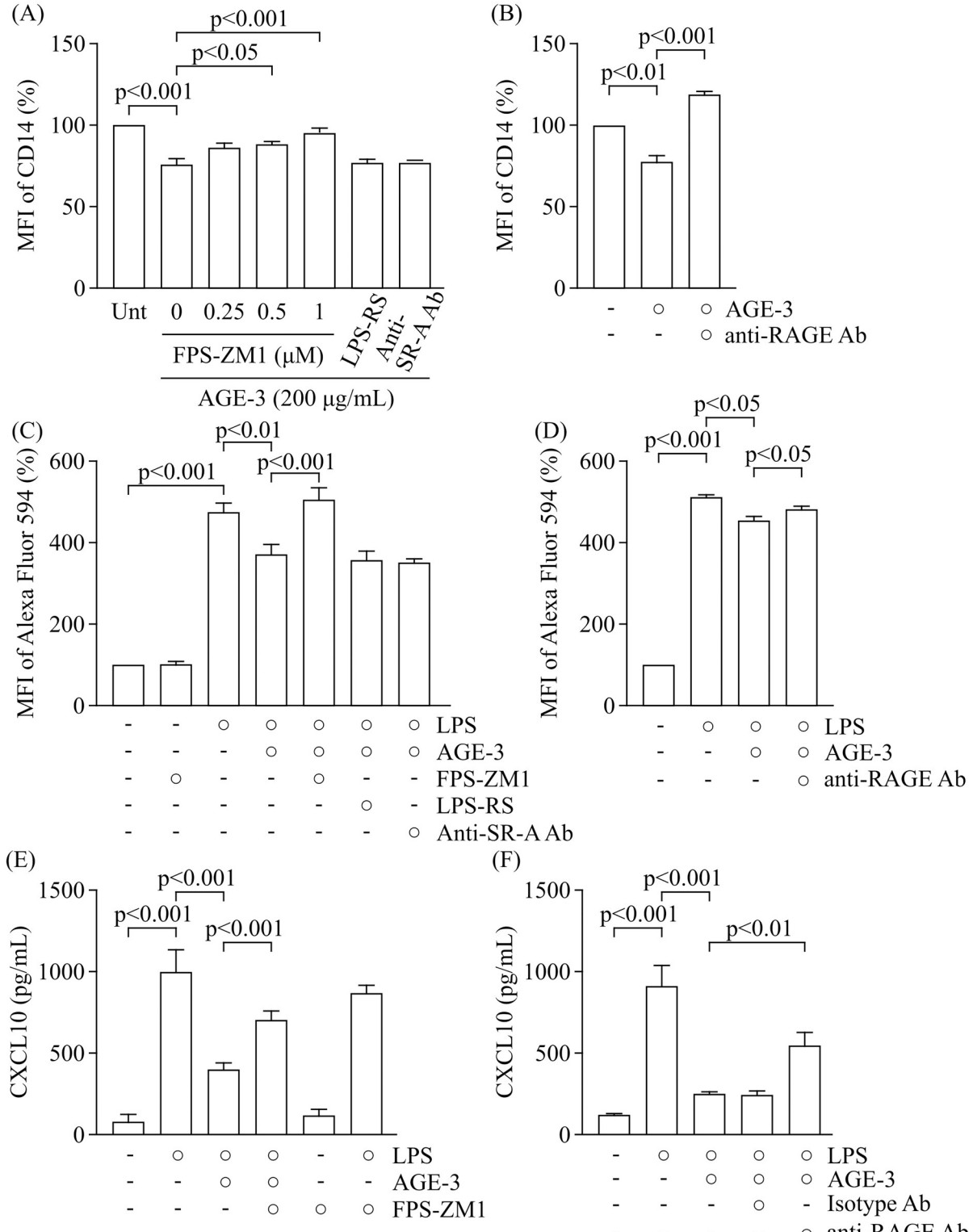

**Fig 4. Involvement of RAGE on the inhibitory effect of AGE-3.** (A) Effect of pharmacological inhibitors on the AGE-3-induced downregulation of CD14 expression (n = 3; AGE-3+LPS-RS and AGE-3+Anti-SR-A Ab, n = 6; Unt, AGE-3 and AGE-3+FPS-ZM1 at 0.25–1.0 µM, Tukey's test). Cells were concomitantly treated with FPZ-ZM1 at 1 µM or LPS-RS at 10 µg/mL and AGE-3 at 200 µg/mL for 4 h. (B) Effect of anti-RAGE neutralising antibody on the AGE-3-induced downregulation of CD14 expression (n = 3). Cells were pre-treated with anti-RAGE antibody at 20 µg/mL for 1 h before incubation with AGE-3 for 4 h. (C) Effect of FPS-ZM1 on LPS uptake in RAW264.7 cells

treated with AGE-3 (n = 9, Tukey's test). Cells were concomitantly treated with Alexa Fluor 594-labelled LPS at 1 μg/mL, AGE-3 at 200 μg/mL, and/or FPS-ZM1 at 1 μM for 4 h. (D) Effect of anti-RAGE neutralising antibody on LPS uptake in RAW264.7 cells. Cells were pre-treated with anti-RAGE antibody at 20 μg/mL for 1 h before incubation with Alexa Fluor 594-labelled LPS and AGE-3 for 4 h. MFI relative to the untreated group (medium alone) was arbitrarily defined as 100%. (E) Effect of FPS-ZM1 at 1 μM and anti-RAGE neutralising antibody on CXCL10 release from RAW264.7 cells treated with LPS at 1 μg/mL and/or AGE-3 at 200 μg/mL (n = 8, Tukey's test). CXCL10 levels in the culture supernatants obtained from cells treated with LPS, AGE-3, FPS-ZM1 and/or anti-RAGE neutralising antibody for 24 h were measured by ELISA. (F) Effect of anti-RAGE neutralising antibody on CXCL10 release from RAW264.7 cells treated with LPS at 1 μg/mL and/or AGE-3 at 200 μg/mL (n = 5, Tukey's test). CXCL10 levels in the culture supernatants obtained from cells treated with LPS, AGE-3 and/or anti-RAGE neutralising antibody for 24 h were measured by ELISA. Data are presented as the means ± SEM.Unt; untreated. Ab; antibody.

AGE-3 on CXCL10 levels was much (62% suppression). These results indicate that the effect of AGE-3 on the innate immune reaction in response to LPS have multiple mechanisms of action including inhibition of LPS uptake. In the present study, we found that AGE-3-induced impaired innate immune reaction in response to LPS is mediated, at least in part, by inhibition of LPS uptake. CD14 is a molecule critical for the first step in eliciting innate immune responses because it is necessary for LPS uptake and the activation of TLR4 signalling pathways [40, 41]. Although CD14 may be involved in monomeric LPS recognition and transfer to a coreceptor or directly into the plasma membrane in myeloid cells, the intracellular trafficking of LPS to the Golgi is independent of CD14 [42, 43]. LPS aggregates are likely to be transported in macrophages to lysosomes in conjunction with CD14, where acyloxyacyl hydrolase deacylates LPS. The latter function is more relevant to the detoxification and clearance of endotoxin rather than the initiation of signalling. Consistent with previous reports using flow cytometry [44, 45], we found that CD14 expression on macrophages was decreased to a minimum level at 1 h after LPS treatment, and then gradually returned to basal levels up to 4 h later. In general, LPS activates TRIF-dependent signalling and facilitates the transcription of CXCL10 mediated by the phosphorylation of IRF3. Previous report has demonstrated that upregulation of CXCL10 mRNA by LPS began to increase and reached maximal level at 4 h in peritoneal macrophage prepared from mouse [27]. And then CXCL10 protein are produced by translation and released into culture medium. Therefore, we suggest that the time lag is occurred between CD14-initiated LPS uptake and CXCL10 production. Additionally, LPS activate release other proinflammatory cytokines such as TNF-α in macrophage. The TNF-α release from macrophage cells occur within a short period (<4hr) after exposure to LPS [46, 47]. TNF-α also increases CXCL10 production in macrophage [48]. It raises possibility that TNF-α released from RAW264.7 cells treated with LPS synergically induce CXCL10 release after long-term exposure to LPS. CD14 was involved in the LPS uptake and CXCL10 production in RAW264.7 cells, indicating that CD14-initiated LPS uptake activated RAW264.7 cells. In the present study, our results suggest that AGEs-induce downregulation of CD14 expression is partially involved in reduction of CXCL10 release in RAW264.7 cells treated with LPS. LPS decreases CD14 expression on cell surface, since LPS binding to CD14 is endocytosed into cell. Although uptake of AGEs is occurred in RAW264.7 cells, anti-CD14 antibody did not affect AGEs uptake, indicating that CD14 independent mechanisms contribute to AGEs uptake. Our previous report has demonstrated that SR-A is associated with AGEs uptake in RAW264.7 cells [15]. It has been reported that stimulation of monocyte by activating agents such as LPS and IFN-γ results in downregulation of CD14 expression on cell surface mediated by shedding of CD14 [49]. CD14 is anchored to the cell membrane by glycosylphosphatidylinositol-anchor [50]. Protease including matrix metalloproteinases (MMPs) contribute to the proteolytic shedding of CD14 [51]. AGE/RAGE signaling increases expression and activity of MMP-9 in macrophage [52]. Therefore, one possibility is that AGE/RAGE decreases CD14 expression on cell surface through excessive shedding of CD14, resulting in decreased CXCL10 production after exposure to LPS.

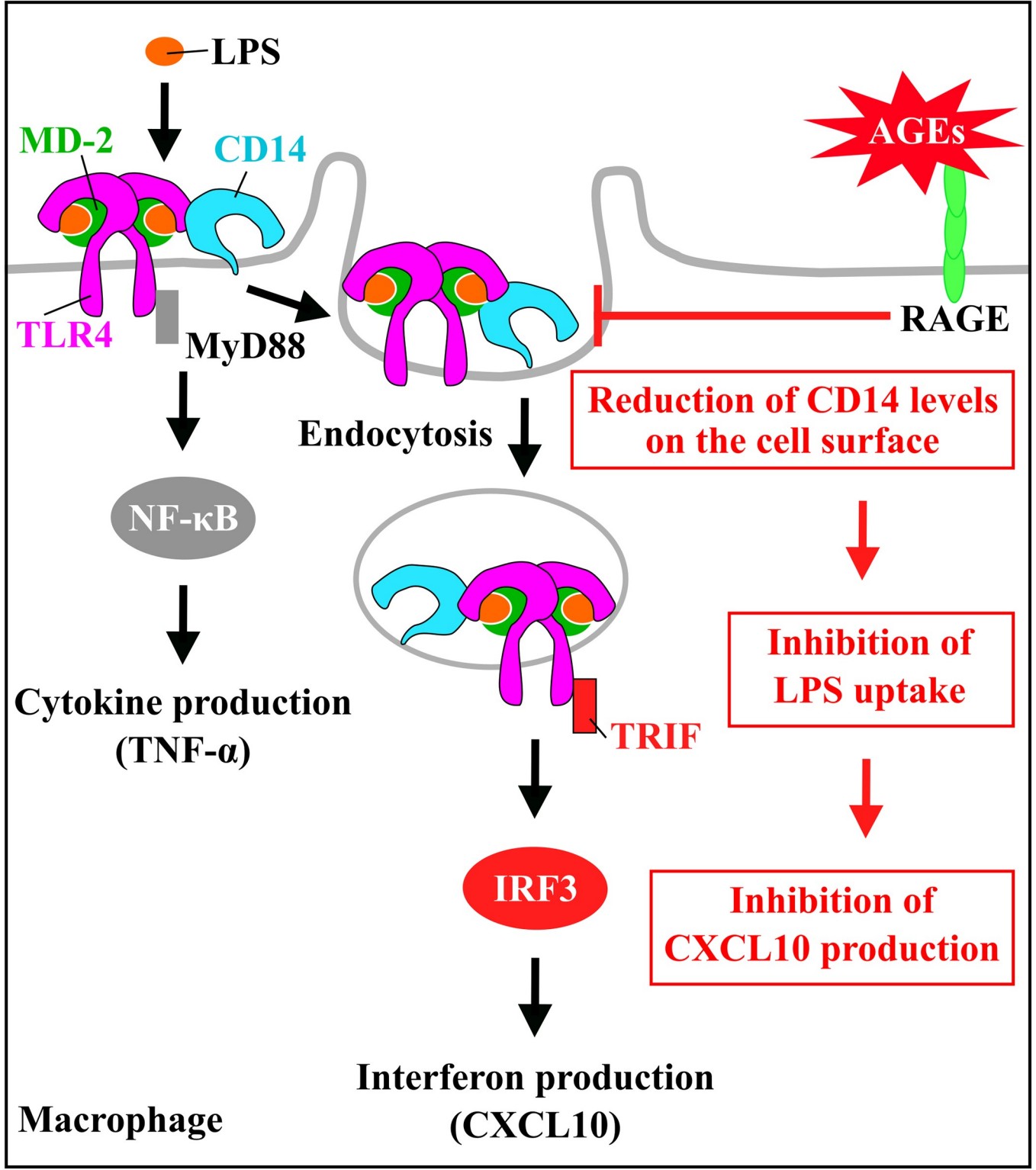

**Fig 5. Model depicting the inhibitory effect of AGE-3 on the response to LPS in macrophages.** LPS binding to CD14 is endocytosed via the formation of a dimerised TLR4/MD-2 complex. Subsequently, this assembly activates TRIF signalling on endosomes and the transcription of IFN-inducible genes including CXCL10. AGE-3 induces the downregulation of CD14 expression though RAGE. Downregulation of CD14 expression induced by AGE-3 contributes to the decreased LPS uptake and CXCL10 release in macrophages.

Among the factors known to undermine host defence, diabetes mellitus increases the susceptibility of patients to viral, bacterial, and fungal infections mainly by modulating the immune system [53]. Therapeutic strategies targeting the innate immune system are important for the treatment of diabetic disorders. Acute hyperglycaemia significantly altered innate immune responses to infection, causing poor outcomes in hospitalised hyperglycaemic patients [53]. High glucose concentrations alone may alter the production of immune mediators in RAW264.7 cells [54]. Although control of the serum sugar level is an effective treatment strategy for infections in patients with diabetes, measures to prevent immune modulation induced by toxic AGEs have not been fully considered.

Our study has a number of limitations. The first limitation is that we have investigated the LPS uptake by using flow cytometry until 4 h after exposure to LPS. When RAW264.7 cells were incubated with LPS for more than 4 h, the analysis could not be accurately performed, because distribution of cell population was confused. This phenomenon might mean cell damage induced by LPS. Although the cell viability measured by WST-8, which is a water-soluble tetrazolium salt and used for assessing cell metabolic activity, did not alter in RAW264.7 cells treated with LPS for 24 h. It is possibility that the manipulation including detachment of cell from plate or centrifugation during the processing of flow cytometry potentially damage cell. In contrast, assessment of cell viability using WST-8 only add detection reagent and is less likely to damage cells. Therefore, we could not assess LPS uptake when cells are exposed to LPS more than 4 h. The second limitation is that the evidence for involvement of RAGE on AGE-induced suppression of response to LPS have not adequately established. In western blot analysis for protein expression of RAGE, indistinct bands were observed in lysate of RAW264.7 cells in contrast to lung tissue. In the present study, we did not confirm the involvement of RAGE using knockout or knockdown experiments. High concentration of FPS-ZM1 at 10 μM has inhibitory effect on androgen receptor, benzodiazepine receptor, kappa opioid receptor and melatonin receptor [55]. However, none of these receptors bind to AGEs. In addition, our results showed that FPS-ZM1 alone did not affect LPS-induced CXCL10 release. Although our results remain limitation, we considered that FPS-ZM1 and anti-RAGE antibody inhibit AGEs though blockage of RAGE in the main. The third limitation is that our findings are obtained by a single murine cell line. Further studies are needed to confirm the inhibitory effect of AGEs on innate immune reaction response to LPS and its molecular mechanisms using human peripheral blood monocyte cell.

In conclusion, we report the mechanism involved in the reduced clearance of LPS and cytokine release by diabetic subjects. The present study indicates that modifying endotoxaemia might be useful for treating complications associated with diabetes. Future studies should continue to elucidate the mechanisms related to the effects of AGE on infectious patients.

## Supporting information

**S1 Fig. Viability of RAW264.7 cells incubated with LPS and/or AGE-3.** Cells were treated with LPS at 1 μg/mL and/or AGE-3 at 200 μg/mL for 24 h (n = 5 means the number of independent experiments). Data are expressed as means ± SEM and are normalised to the untreated group.
(TIF)

**S2 Fig. Effect of LPS on receptor expression on the cell surface of RAW264.7 cells.** Cells were treated with LPS at 1 μg/mL for 24 h and then the expression of each receptor was determined by flow cytometry. (A) SR-A (n = 4 means the number of independent experiments), (B) LOX-1 (n = 3 means the number of independent experiments, Student's *t*-test), (C) RAGE (n = 3 means the number of independent experiments, Student's *t*-test), (D) CD36 (n = 3

means the number of independent experiments), and (E) TLR4 (n = 3 means the number of independent experiments, Student's *t*-test). Data are expressed as means ± SEM and are normalised to the untreated group.
(TIF)

**S3 Fig. Effect of neutralising antibodies on the LPS uptake in RAW264.7 cells.** Cells were pre-treated with each neutralising antibody for 1 h before treatment with Alexa Fluor 594-lebelled LPS at 1 μg/mL for 4 h. The MFI was measured by flow cytometry (n = 3 means the number of independent experiments). MFI; mean fluorescence intensity. Unt; untreated.
(TIF)

**S4 Fig. Effect of isotype control for anti-CD14 neutralising antibody on the LPS uptake in RAW264.7 cells.** Cells were pre-treated with isotype control antibody for 1 h before treatment with Alexa Fluor 594-lebelled LPS at 1 μg/mL for 4 h. The MFI was measured by flow cytometry (n = 3 means the number of independent experiments). MFI; mean fluorescence intensity. Unt; untreated.
(TIF)

**S5 Fig. Effect of LPS and neutralising antibodies against SR-A or CD14 on AGE uptake in RAW264.7 cells.** (A) Cells were pre-treated with anti-SR-A antibody for 1 h before treatment with Alexa Fluor 488-labelled AGE-3 at 200 μg/mL. LPS at 1 μg/mL was added concomitant with fluorescent-labelled AGE-3. At 1 h after Alexa Fluor 488-labelled AGE-3 treatment, the MFI was measured by flow cytometry (n = 3 means the number of independent experiments, Turkey's test). (B) Cells were pre-treated with anti-CD14 antibody for 1 h before treatment with Alexa Fluor 488-labelled AGE-3 at 200 μg/mL. At 1 h after Alexa Fluor 488-labelled AGE-3 treatment, the MFI was measured by flow cytometry (n = 3 means the number of independent experiments, Turkey's test). MFI relative to the untreated group (medium alone) was arbitrarily defined as 100%. Data are expressed as means ± SEM. MFI; mean fluorescence intensity. Unt; untreated.
(TIF)

**S6 Fig. Protein expression of RAGE in RAW264.7 cells and mouse lung tissue.** RAGE expression was assessed by using western blot. Representative western blot demonstrating RAGE expression in RAW264.7 cells and mouse lung tissue. The two bands in the vicinity of 50 kDa represent the pre and post glycation type of RAGE protein. GAPDH used as loading control.
(TIF)

# Acknowledgments

The authors would like to thank the staff at the Central Research Facilities, Kindai University Faculty of Medicine, Center for Instrumental Analyses and Center for Morphological Analyses for their technical assistance. We thank J. Ludovic Croxford, PhD, from Edanz Group (https://en-author-services.edanzgroup.com/ac) for editing a draft of this manuscript.

# Author Contributions

**Conceptualization:** Atsuhiro Kitaura, Takashi Nishinaka, Hidenori Wake, Shuji Mori, Shinichi Nakao, Hideo Takahashi.

**Data curation:** Atsuhiro Kitaura, Takashi Nishinaka, Shinichi Hamasaki, Omer Faruk Hatipoglu.

**Formal analysis:** Atsuhiro Kitaura, Takashi Nishinaka, Omer Faruk Hatipoglu.

**Funding acquisition:** Atsuhiro Kitaura, Masahiro Nishibori, Shinichi Nakao, Hideo Takahashi.

**Investigation:** Atsuhiro Kitaura, Takashi Nishinaka, Omer Faruk Hatipoglu.

**Methodology:** Atsuhiro Kitaura, Takashi Nishinaka, Shinichi Hamasaki, Shuji Mori.

**Project administration:** Atsuhiro Kitaura, Hideo Takahashi.

**Resources:** Hidenori Wake, Masahiro Nishibori, Shuji Mori.

**Supervision:** Hideo Takahashi.

**Validation:** Atsuhiro Kitaura, Takashi Nishinaka, Shinichi Hamasaki, Omer Faruk Hatipoglu.

**Visualization:** Atsuhiro Kitaura, Takashi Nishinaka.

**Writing – original draft:** Atsuhiro Kitaura, Takashi Nishinaka, Hideo Takahashi.

**Writing – review & editing:** Atsuhiro Kitaura, Takashi Nishinaka, Shinichi Hamasaki, Omer Faruk Hatipoglu, Hidenori Wake, Masahiro Nishibori, Shuji Mori, Shinichi Nakao, Hideo Takahashi.

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
