## [Decision Letter · Decision Letter 0]

18 Sep 2020

PONE-D-20-26201

Advanced glycation end-products reduce lipopolysaccharide uptake by macrophages

PLOS ONE

Dear Dr. Takahashi,

Thank you for submitting your manuscript to PLOS ONE. After careful consideration, we feel that it has merit but does not fully meet PLOS ONE’s publication criteria as it currently stands. Therefore, we invite you to submit a revised version of the manuscript that addresses the points raised during the review process.

We look forward to receiving your revised manuscript.

Kind regards,

David M. Ojcius

Academic Editor

PLOS ONE

Journal Requirements:

2. Please ensure you have provided the source, catalog number, and final concentration of all neutralizing antibodies, isotype negative control antibodies, and secondary antibodies.

Please also provide the catalog number of the RAW264.7 cells.

Reviewers' comments:

Reviewer's Responses to Questions

**Comments to the Author**

1. Is the manuscript technically sound, and do the data support the conclusions?

Reviewer #1: Yes

Reviewer #2: No

Reviewer #3: Partly

2. Has the statistical analysis been performed appropriately and rigorously? 

Reviewer #1: Yes

Reviewer #2: Yes

Reviewer #3: I Don't Know

3. Have the authors made all data underlying the findings in their manuscript fully available?

Reviewer #1: Yes

Reviewer #2: Yes

Reviewer #3: No

4. Is the manuscript presented in an intelligible fashion and written in standard English?

Reviewer #1: Yes

Reviewer #2: Yes

Reviewer #3: Yes

5. Review Comments to the Author

Reviewer #1: The authors describe results from a study that elucidated how the advanced glycation end products AGE-3 inhibits LPS uptake by cultured macrophage cells. By preventing uptake, AGE-3 prevents its removal from the circulation and detoxification.

-Page 6, line 11: What is meant by ‘cross tolerance between AGE and LPS’?

-Do the sample sizes in the figures represent the number of wells/plates? Or unique batches of cells?

-Figure 2 and 3 legend: be sure to define the abbreviation MFI.

-Figure 3F is interesting. Why do you suppose that AGE-3 inhibits LPS uptake acutely (1 and 2 hours) but not as well at 4 hours?

-Page 25, lines 11-12: Do you mean that the RAGE antagonist prevented the AGE-3 decline in CD14? It is difficult to interpret what you meant by restored downregulation – in my mind that means it helped downregulate CD14.

-Page 25, line 16: I did not see a supplemental figure #4.

-Lines 8-9, page 34: It is mentioned that RAGE was not detected using a commercial antibody indicating low expression. Was the antibody validated? Is it possible the antibody was not specific resulting in the low detection?

-It is not necessary to refer to figure numbers in the discussion section.

-Please add a hypothesis to the introduction and abstract.

Reviewer #2: This manuscript presents data supporting the conclusion that AGE, working through RAGE, downregulates CD14, thus decreasing LPS uptake and signaling as measured by CX CL10 release. While the results are interesting, they are too preliminary as most of the results are incremental increases or repeats of what is already known. The results are descriptive with no mechanism presented or tested as to how AGE, working through RAGE or SR-A, signal to down-regulate CD14. It is not surprising that down-regulation of CD14 would decrease LPS uptake and it has already been shown that, in monocytes, CD14 is linked to CXCL10 expression. As such, it is this reviewer’s opinion that the data here are too preliminary for publication at this time.

Figure 1 sets up the system, although the conclusion that at 4 hr LPS uptake had “reached maximal level” is misleading as there are no later timepoints to suggest that uptake decreases after that 4 hr (or that it doesn’t continue to rise).

Figure 2. Demonstrates that LPS internalization results in loss of CD14 from the membrane. While internalization of CD14 with LPS may not have been shown before in exactly this way, it is not surprising and thus, represents an incremental increase in our knowledge. Similarly, the ability of CD14 Ab to decrease CXCL10 is derivative of the previous data showing that only CD14+ monocytes release CXCL10 in response to LPS (PMID: 19901067). How do the authors link the results at 4 hr (CD14, LPS uptake) (Figs 2 A,B) to CXCL10 release in cells treated with LPS and Ab for 24 h (Fig 2C)?

The novelty of this work comes from the impact that AGE have on LPS signaling. AGE reduce LPS uptake although the decrease even at the max does of AGE is modest (although statistically significant) and raises the question of physiological relevance. The representative flow in Fig 3A calls into question whether they is much LPS at all being taken up as there is significant overlap of LPS, LPS+AGE is the untreated and the MFI is so very low. Is the timing right? The longest time used is 4 h; the authors concluded that 4 h was maximal but did not show later timepoints to establish that 4 h represents a plateau of uptake. However, when looking at CXCL10, they incubated the cells with AGE and LPS for 24 h. How do the authors conclude that what happens by 4 h impacts CXCL10 at 24 h in light of the fact that CD14 levels apparently return to normal by 4 h (Fig 2A).

Fig 4. Same comment about timing of CXCL10. All of the results could be predicted from previous published work (that AGE works through RAGE, that AGE decreases LPS uptake and CD14 expression). The novelty is the CXCL10 results but they are done differently than all the other expeirments in the paper so there is not direct link between AGE-CD14-CXCL10. While a cartoon is presented, the critical question remains unanswered: HOW does AGE/RAGE cross-talk with LPS/CD14 to reduce CXCL10 signaling? As such, this body of work is preliminary and needs a more in depth investigation of the signaling network underlying the descriptive results presented.

Statistics: RAW 264.7 is a cell line. Thus, one would expect minimal variation in the results, making n=3 acceptable for the number of experiments run. While a presentation of the variation in 3 points as SEM is published, it is a bit misleading, especially given the very small SEM bars on Fig 3F and Fig 4.

Minor comments:

English in places could be improved

• Figure legend 1 “fluorescence-labelled LPS”, Alex 594-LPS would be better.

• CD14 “is not indispensable for” is a double negative. “CD14 is necessary for” would be clearer.

Fig legend for Fig 3E; lowest concentration of 0.2 as in legend or 2.0 as in bar graph/text?

Figs 3-4 represent the novelty. Many of the controls (ie blocking of other receptors doesn’t alter LPS uptake) could be shown.

Reviewer #3: The manuscript by Kitaura et al. presents the interdependent effects of AGE3 (a particular type of advanced glycation end-product), lipopolysaccharide (LPS) and the role of the RAGE receptor for the activation of murine macrophages. The authors conclude that RAGE activation by AGE3 impairs immune response in diabetic patients.

The described experiments appear to be done carefully and support to overall conclusion of the manuscript. Weak points that should be addressed are the reliance on FACS data for AGE3 and LPS uptake, as well as for cell surface receptor expression. This could be remediated with additional experiments (see details listed below), including additional controls and alternate methodologies (e.g. qPCR / Western Blots for receptor transcription and expression and confocal microscopy for receptor localization. There are some minor correction necessary in the Material / Methods / Experimental sections (details below).

The discussion should more clearly point out the relevance of AGE 3 in diabetic patients and the limitations of translating results from cell based assays using mouse cells to human pathology and therapy.

The authors should provide additional detail regarding the following:

Materials and Methods:

Q1: Page 8 / line 8: What was the concentration of glycoaldehyde used to form AGE-3. How was excess glycoaldehyde removed at the end of the incubation period?

Q2: AGE 8 line 1: What does the acronym SRL stands for?

Q3: PAGE 8: the list of antibodies used in incomplete. See page 13 for other antibodies used.

Q4: PAGE 10 line 1: What was the final labeling stoichiometry of the Alexa fluorophore bound to AGE-3? BSA has only one free SH group. Is the SH group still accessible after glycation?

Q5: PAGE10 line 7/8: What does “the strength of Alexa fluor 488-BSA or AGE 3 per unit dosage were adjusted” mean? That may refer in part to Q4

Q6: PAGE 11 line 1: Please identify and list all neutralizing antibodies in “Materials”

Q7: PAGE 12 line1: How were cells detached (mechanically or chemically) for FACS analysis

Q8: PAGE 12 line 7 LS-C212626 and RAGE Ab appears to be a polyclonal rabbit anit0human RAGE Ab. How do the authors know that (1) the Ab recognizes mouse RAGE (2) is specific for RAGE (3) shows no cross-reactivity with other cellular proteins. This is particular concerning given that RAGE is apparently expressed at so low levels that it cannot be detected by Western blot.

Statistical Analysis

Q9: The FACS data a generally scaled to a control set to 100% and values for treatments are reported % relative to controls. That means the controls have by default no error or deviation (all 100%). Doing data analysis this way can really cover up large differences between biological independent repeats. It is also not clear if any independent repeats were done. If no independent repeats were done, then scaling to 100% might be okay because then all replicates are just technical replicates. The authors should clarify this point. They should also justify the use of specific post-hoc test, under consideration that controls (defaulted to 100%) have no error.

Results:

Q10: Figure 1 (A) Explain why the untreated cells show two distinct populations, i.e. two FACS peaks.

Q11: Figure 1 (A): Please label both axes of the FACS figure, in particular the fluorescence intensity. That will allow the reader to judge the magnitude of MFI changes shown in the other panels of figure 1.

Q12: Again FACS data are shown as scaled to 100% controls. That makes it impossible to actually judge the sale of treatment related changes. This in turn makes it hard to imply that statistical significance could have biological relevance. While difference s between treatment groups appear to be of up to more than three-fold change, providing actual quantitative data would strengthen the data analysis.

Q13: Data interpretation in the written results section is mostly qualitative in nature. It strengthen the paper if the authors would discuss data on a quantitative level.

Q14: Figure 3(A) same as Q11, figures need axes labels

Q15: PAGE 22 line 15: Please provide a more detailed description of how the AGE3 IC50 for LPS uptake was calculated. The four values extractable from figure 3(B) are not sufficient to make such a calculation. Especially since there are not data that justify the statment “ we assumed that the effects …were maximal at 200 ug/ml”. If no additional data are available, the IC50 statement should be removed.

Q16: Figure 3(F): AGE 3 up to 20 ug/ml has no effect on CD14 cell surface level in the presence of LPS (Figure 3F is basically similar to figure 2A). 2 or 20 ug/ml AGE 3 also has no effect on CD14 expression (Figure 3E). IN light of this the statement on PAGE 23 line 9-11 seems poorly supported. In stead, down regulation of CXCL10 could be attributed to AGE3 alone and be independent of CD14.

The following control experiment should be done: measure the effect of AGE 3 on CXCL10 (that would be an additional entry in Figure 4C)

Q17: PAGE 25 line 11/12: what is known about the target specificity / off-target effects of FPS-ZM1. Small molecule drugs are notorious for off-target effects. To confirm RAGE involvement additional experiments with monoclonal anti-RAGE antibodies should be done. In essence, the data shown in figures 4A, B and C should include an neutralizing anti-RAGE mAb, including a dose response curve (like 4A).

Q18: PAGE 28 line 9/10: What are the absolute RAGE expression levels on RAW624.7 cells? AS mentioned, the anti-RAGE Ab used for FACS seems to be inappropriate (polyclonal, anti-human RAGE). The authors should employ additional confirmatory experiments to show RAGE expression on RAW cells. Why would a Western blot not work when RAGE is there is significant quantities? If WB does not work, show IF micrographs and do an ELISA. FACS as the only method is not convincing.

Q19: Discussion: How do the results obtained from the cultured mouse cells translate to humans with diabetes? Discuss the limitations.

Q20 Discussion: What is the AGE3 (not just “AGE” in general) level in diabetic vs. healthy patients? How does that compare to the 200 ug/ml used in most experiments?

Q21: Figure 5: The figure could suggest that AGE3/RAGE down-regulate CD14 on the genetic level (CD14 expression down

6. PLOS authors have the option to publish the peer review history of their article (what does this mean?). If published, this will include your full peer review and any attached files.

Reviewer #1: No

Reviewer #2: No

Reviewer #3: No

---

## [Author Response · Author response to Decision Letter 0]

26 Oct 2020

Response to reviewers:

Response to the requirements:

 We have revised the file name. We have ensured PLOS ONE’s style requirements. 

2. Please ensure you have provided the source, catalog number, and final concentration of all neutralizing antibodies, isotype negative control antibodies, and secondary antibodies.

Please also provide the catalog number of the RAW264.7 cells.

Response to the requirements:

 We have added the information of cell line. RAW264.7 (EC91062702, DS Pharma Biomedical, Osaka, Japan) (Page 10, Line 4). 

Response to the requirements:

 Thank you for valuable suggestions. We have deleted the phrase “data not shown” in our manuscript. The essential data were included in main body of manuscript or supporting information. 

4. In your Data Availability statement, you have not specified where the minimal data set underlying the results described in your manuscript can be found. PLOS defines a study's minimal data set as the underlying data used to reach the conclusions drawn in the manuscript and any additional data required to replicate the reported study findings in their entirety. All PLOS journals require that the minimal data set be made fully available. For more information about our data policy, please see http://journals.plos.org/plosone/s/data-availability. Upon re-submitting your revised manuscript, please upload your study’s minimal underlying data set as either Supporting Information files or to a stable, public repository and include the relevant URLs, DOIs, or accession numbers within your revised cover letter. For a list of acceptable repositories, please see http://journals.plos.org/plosone/s/data-availability#loc-recommended-repositories. Any potentially identifying patient information must be fully anonymized.

Important: If there are ethical or legal restrictions to sharing your data publicly, please explain these restrictions in detail. Please see our guidelines for more information on what we consider unacceptable restrictions to publicly sharing data: http://journals.plos.org/plosone/s/data-availability#loc-unacceptable-data-access-restrictions. Note that it is not acceptable for the authors to be the sole named individuals responsible for ensuring data access. We will update your Data Availability statement to reflect the information you provide in your cover letter.

Response to the requirements:

 Thank you for valuable suggestions. The minimal data set were included in main body of manuscript or supporting information.

 

Response to reviewers:

Reviewer #1

We would like to thank reviewer #1 for the valuable comments and suggestions on our manuscript. We have carefully read the comments and have made the following corrections in the revised version:

1. Page 6, line 11: What is meant by ‘cross tolerance between AGE and LPS’?

Reply to the comment 1:

 We have added the explanation about cross tolerance between AGE and LPS as follows (Page 6, Line 11-12).

Cross tolerance, which priming of macrophage with AGE show decreased responses to LPS, has been established [16], indicating that AGEs is involved in impaired immune responses to microbes.

2. Do the sample sizes in the figures represent the number of wells/plates? Or unique batches of cells?

Reply to the comment 2:

In the present study, sample size means independent experiments. We have added the explanation of sample size in each figure legends.

3. Figure 2 and 3 legend: be sure to define the abbreviation MFI.

Reply to the comment 3:

 We have added the abbreviation of MFI in “Material and Methods” and each figure legends as follow. MFI; mean fluorescence intensity. 

4. Figure 3F is interesting. Why do you suppose that AGE-3 inhibits LPS uptake acutely (1 and 2 hours) but not as well at 4 hours?

Reply to the comment 4:

 You have raised an important question. The reason why AGE-3 acutely inhibits LPS uptake is unknown. We have added the discussion for acute effect of AGE-3 on LPS uptake and CD14 expression (Page 32, Line 16-Page 33, Line 2). 

 Furthermore, different effect of AGEs in the presence or absence of LPS may contribute to the acute inhibitory effect of AGEs on LPS uptake and CD14 expression. 

5. Page 25, lines 11-12: Do you mean that the RAGE antagonist prevented the AGE-3 decline in CD14? It is difficult to interpret what you meant by restored downregulation – in my mind that means it helped downregulate CD14.

Reply to the comment 5:

 Thank you for your suggestion. Our result indicated that RAGE antagonist prevents AGE-3-induced downregulation of CD14 expression. As you pointed out, we have revised sentence as follows (Page 28 line 4-6).

A RAGE antagonist, FPS-ZM1 at 0.5 and 1 μM, and neutralising antibody against RAGE at 20 μg/mL prevented the AGE-3-induced downregulation of CD14 expression (Figs 4A and 4B).

6. Page 25, line 16: I did not see a supplemental figure #4.

Reply to the comment 6:

Thank you for pointing out the mistake. We have revised and added the supplementary figure.

S1 Fig. Viability of RAW264.7 cells incubated with LPS and/or AGE-3.

S2 Fig. Effect of LPS on receptor expression on the cell surface of RAW264.7 cells.

S3 Fig. Effect of neutralising antibodies on the LPS uptake in RAW264.7 cells.

S4 Fig. Effect of isotype control for anti-CD14 neutralising antibody on the LPS uptake in RAW264.7 cells.

S5 Fig. Effect of LPS and neutralising antibodies against SR-A or CD14 on AGE uptake in RAW264.7 cells.

S6 Fig. Protein expression of RAGE in RAW264.7 cells and mouse lung tissue.

7. Lines 8-9, page 34: It is mentioned that RAGE was not detected using a commercial antibody indicating low expression. Was the antibody validated? Is it possible the antibody was not specific resulting in the low detection?

Reply to the comment 7:

 Thank you for comment. We verified specificity and reactivity on commercially available anti-RAGE antibodies (Abcam, #ab3611) by using western blot. The #ab3611 has detected the two bands, which represent pre and post glycation type of RAGE protein, in the vicinity of 50 kDa in mouse lung tissue as positive control. The protein sample prepared from RAW264.7 cells showed weak band in the vicinity of 50 kDa, indicating that RAW264.7 cell shows much lower levels of RAGE compared to lung tissue. Therefore, commercially available anti-RAGE antibodies work to detect RAGE protein. Niven et al reported that RAGE expression of RAW264.7 cells have confirmed at mRNA and protein levels [33]. We suggest that RAGE is a functional receptor for AGEs despite expression are low in RAW264.7 cells. 

 We have added the materials and methods and the result which indicate much lower protein expression of RAGE in RAW264.7 cells (S6 Fig). We have also revised the results and discussion.

(Results, Page 28, Line 13- Page 29, Line 5)

 We verified the protein expression of RAGE in RAW264.7 cells by using western blot. Consistent with previous study [33], the two bands, which represent pre and post glycation type of RAGE protein, in the vicinity of 50 kDa were observed in mouse lung tissue used as positive control (S6 Fig). The protein sample prepared from RAW264.7 cells showed weak band in the vicinity of 50 kDa, indicating that RAGE is a functional receptor for AGEs despite expression are low in RAW264.7 cells. Taken together, these findings suggest that AGE-3 attenuated the response to LPS in RAW264.7 cells, which was partially mediated by the activation of RAGE (Fig 5).

(Discussion, Page 32, Line 10-11)

 We detected low protein expression of RAGE in RAW264.7 cells.

Reference

33. Niven J, Hoare J, McGowan D, Devarajan G, Itohara S, Gannagé M, et al. S100B Up-Regulates Macrophage Production of IL1β and CCL22 and Influences Severity of Retinal Inflammation. Lewin AS, editor. PLoS One. 2015;10: e0132688. doi:10.1371/journal.pone.0132688

8. It is not necessary to refer to figure numbers in the discussion section.

Reply to the comment 8:

As you pointed out, we have deleted figure numbers in the discussion section.

9. Please add a hypothesis to the introduction and abstract.

Reply to the comment 9:

 We appreciate your valuable comment. As you pointed out, we have added a hypothesis related to our study in the introduction and abstract as follows. 

(Abstract, Page 2, Line 13-15)

 Previously, we reported that AGE-3 was internalised into RAW264.7 cells through scavenger receptor-1 Class A. We hypothesized that AGE uptake interrupt LPS uptake and impair innate immune reaction response to LPS in RAW264.7 cells. 

(Introduction, Page 6, Line 14-15)

We hypothesized that AGE uptake interrupt LPS uptake and impair innate immune reaction response to LPS in macrophage. 

 

Reviewer #2

We would like to thank reviewer #2 for the valuable comments and suggestions on our manuscript. We have carefully read the comments and have made the following corrections in the revised version:

1. This manuscript presents data supporting the conclusion that AGE, working through RAGE, downregulates CD14, thus decreasing LPS uptake and signaling as measured by CX CL10 release. While the results are interesting, they are too preliminary as most of the results are incremental increases or repeats of what is already known. The results are descriptive with no mechanism presented or tested as to how AGE, working through RAGE or SR-A, signal to down-regulate CD14. It is not surprising that down-regulation of CD14 would decrease LPS uptake and it has already been shown that, in monocytes, CD14 is linked to CXCL10 expression. As such, it is this reviewer’s opinion that the data here are too preliminary for publication at this time.

Reply to the comment 1:

 In the present study, we found that AGE suppressed LPS uptake and CXCL10 production. AGE-induced suppressive effects on the reaction response to LPS were mediated at least in part by reduction of CD14 on the cell surface. Furthermore, blockage of RAGE restored AGE-induced suppressive effects. We believe that our findings also us to propose an importance of AGE-RAGE signaling on impaired innate immune reaction in patients with diabetes. As you pointed out, it is unclear how AGE-RAGE signaling have negatively effects on LPS-CD14 pathway leading to CXCL10 production. Further studies need to investigate molecular mechanisms underlying the effect of AGE-RAGE signaling on impaired innate immune reaction.

2. Figure 1 sets up the system, although the conclusion that at 4 hr LPS uptake had “reached maximal level” is misleading as there are no later timepoints to suggest that uptake decreases after that 4 hr (or that it doesn’t continue to rise).

Reply to the comment 2:

 Thank you for valuable comment. In the present study, we have investigated the LPS uptake by using flow cytometry. When RAW264.7 cells were incubated with LPS for more than 4 h, the analysis could not be accurately performed, because distribution of cell population was confused. This phenomenon might mean cell damage induced by LPS. Although the cell viability measured by WST-8, which is a water-soluble tetrazolium salt and used for assessing cell metabolic activity, did not alter in RAW264.7 cells treated with LPS for 24 h. It is possibility that the manipulation including detachment of cell from plate or centrifugation during the processing of flow cytometry potentially damage cell. In contrast, assessment of cell viability using WST-8 only add detection reagent and is less likely to damage cells. Therefore, we could not assess LPS uptake when cells are exposed to LPS more than 4 h. As you pointed out, it is unclear whether LPS uptake by RAW264.7 cells reach maximal levels at 4 hr.

We have added the discussion about limitation of our study as follows (Page 35, Line 4-14). 

Our study has a number of limitations. The first limitation is that we have investigated the LPS uptake by using flow cytometry until 4 h after exposure to LPS. When RAW264.7 cells were incubated with LPS for more than 4 h, the analysis could not be accurately performed, because distribution of cell population was confused. This phenomenon might mean cell damage induced by LPS. Although the cell viability measured by WST-8, which is a water-soluble tetrazolium salt and used for assessing cell metabolic activity, did not alter in RAW264.7 cells treated with LPS for 24 h. It is possibility that the manipulation including detachment of cell from plate or centrifugation during the processing of flow cytometry potentially damage cell. In contrast, assessment of cell viability using WST-8 only add detection reagent and is less likely to damage cells. Therefore, we could not assess LPS uptake when cells are exposed to LPS more than 4 h.

We have revised the results related to time course of LPS uptake as follows (Page 19, Line 10-11). 

The mean florescence intensity (MFI) indicating uptake of fluorescently-labelled LPS at 1 �g/ml increased at 2 h (268%) and 4 h (350%).

3. Figure 2. Demonstrates that LPS internalization results in loss of CD14 from the membrane. While internalization of CD14 with LPS may not have been shown before in exactly this way, it is not surprising and thus, represents an incremental increase in our knowledge. Similarly, the ability of CD14 Ab to decrease CXCL10 is derivative of the previous data showing that only CD14+ monocytes release CXCL10 in response to LPS (PMID: 19901067).

Reply to the comment 3:

In the present study, we found that AGE-3 suppressed LPS uptake in RAW264.7 cells. To determine the effect of AGE-3 on the cell function in RAW264.7 cells related to LPS uptake, we measured several cytokines, which are reported to be upregulated by internalization of LPS, in cell culture supernatant of RAW264.7 cells. We were able to detect CXCL10. As you pointed out, previous reports have demonstrated that neutralizing antibody against CD14 inhibit TNF-α and IL-6 release from macrophage treated with LPS. Our results relating to Fig1 and Fig2 are positioned as the findings to confirm the effect of LPS uptake in RAW264.7. 

4. How do the authors link the results at 4 hr (CD14, LPS uptake) (Figs 2 A,B) to CXCL10 release in cells treated with LPS and Ab for 24 h (Fig 2C)?

Reply to the comment 4:

 As the limitation of our study, we evaluated LPS uptake until 4 h, because cell condition is worse (Please see point “Reply to the comment 2” above). We have added the Fig.2C to demonstrate time course of CXCL10 production after LPS treatment. The remarkably increased CXCL10 production in cell culture supernatant were able to be detected by exposure to LPS for 24 h bot not 4, 8 and 16 h. Previous report has demonstrated that upregulation of CXCL10 mRNA by LPS began to increase and reached maximal level at 4 h in peritoneal macrophage prepared from mouse [26]. And then CXCL10 protein are produced by translation and released into culture medium. Therefore, we suggest that the time lag is occurred between CD14-initiated LPS uptake and CXCL10 production. In the present study, our results possibly suggest that AGE-3-induced suppression of CXCL10 release after exposure to LPS is mediated at least in part by inhibition of LPS uptake via downregulated CD14 expression.

 We have added the figure to show the time course of CXCL10 production in culture supernatants after exposure to LPS (Fig.2C). We have also added the text relating to the time lag between changes in CD14 expression and CXCL10 production as follows (Page 22, Line 7-15). 

 We conformed the remarkably increased CXCL10 production in the culture supernatants were able to be detected by exposure to LPS for 24 h bot not 4, 8 and 16 h (Fig 2C). Previous report have demonstrated that upregulation of CXCL10 mRNA by LPS began to increase and reached maximal level at 4 h in peritoneal macrophage prepared from mouse [26]. And then CXCL10 protein are produced by translation and released into culture medium, indicating that the time lag is occurred between CD14-initiated LPS uptake and CXCL10 production. In experimental condition which is exposed to LPS for 24 h, we found that anti-CD14 antibody at 20 μg/ml also suppressed the increased release of CXCL10 in LPS-treated RAW264.7 cells by 79.5% (Fig 2D).

Reference

26. Tebo JM, Kim HS, Gao J, Armstrong DA, Hamilton TA. Interleukin-10 Suppresses IP-10 Gene Transcription by Inhibiting the Production of Class I Interferon. Blood. 1998;92: 4742–4749. doi:10.1182/blood.V92.12.4742

5. The novelty of this work comes from the impact that AGE have on LPS signaling. AGE reduce LPS uptake although the decrease even at the max does of AGE is modest (although statistically significant) and raises the question of physiological relevance. 

Reply to the comment 5:

 You have raised an important question. Although we have not answered an important question, we have added the discussion based on our idea as follows (Page 33, Line 6-11).

Although AGE-3-induced inhibition of LPS uptake was little (24% inhibition), the suppressive effect of AGE-3 on CXCL10 levels was much (62% suppression). These results indicate that the effect of AGE-3 on the innate immune reaction in response to LPS have multiple mechanisms of action including inhibition of LPS uptake. In the present study, we found that AGE-3-induced impaired innate immune reaction in response to LPS is mediated, at least in part, by inhibition of LPS uptake.

6. The representative flow in Fig 3A calls into question whether they is much LPS at all being taken up as there is significant overlap of LPS, LPS+AGE is the untreated and the MFI is so very low. Is the timing right? 

Reply to the comment 6:

 We have revised Fig.3A. Data altered logarithmic display to biexponential display. Representative histogram shows fluorescence intensity for Alexa Fluor 594-labeled LPS in RAW264.7 cells treated with LPS and AGE-3 for 4 h. The percent of increased fluorescence intensity for Alexa Fluor 594-labelled LPS is 300~350% when untreated data is 100%. Histogram have overlapped each other, since increment of fluorescence intensity is little. The scale of x axis is logarithmic.

7. The longest time used is 4 h; the authors concluded that 4 h was maximal but did not show later timepoints to establish that 4 h represents a plateau of uptake. However, when looking at CXCL10, they incubated the cells with AGE and LPS for 24 h. How do the authors conclude that what happens by 4 h impacts CXCL10 at 24 h in light of the fact that CD14 levels apparently return to normal by 4 h (Fig 2A).

Reply to the comment 7:

 As you pointed out, it is unclear whether LPS uptake by RAW264.7 cells reach maximal levels at 4 hr. Please see point “Reply to the comment 4” above. It is possible that the spending a lot of time is required for LPS-CD14-initiated CXCL10 production due to transcription and translation.

8. Fig 4. Same comment about timing of CXCL10. All of the results could be predicted from previous published work (that AGE works through RAGE, that AGE decreases LPS uptake and CD14 expression). The novelty is the CXCL10 results but they are done differently than all the other expeirments in the paper so there is not direct link between AGE-CD14-CXCL10. While a cartoon is presented, the critical question remains unanswered: HOW does AGE/RAGE cross-talk with LPS/CD14 to reduce CXCL10 signaling? As such, this body of work is preliminary and needs a more in depth investigation of the signaling network underlying the descriptive results presented.

Reply to the comment 8:

 In the present study, for the first time, we found that AGE-3 suppress LPS uptake and CXCL10 levels in response to LPS. These effects are mediated in part by downregulation of CD14. Our findings suggest that impaired innate immune reaction in diabetic patients is associated with increased AGEs and recovered by blockage of AGE-RAGE signaling. As you pointed out, further studies might be needed to clarify the molecular mechanisms regulating CD14 expression by AGE-RAGE signaling.

 Thank you for valuable comment. We have added the discussion about further study as follows (Page 35, Line 15- Page 36, Line 2). 

Further studies are needed to confirm the inhibitory effect of AGEs on innate immune reaction response to LPS and its molecular mechanisms using human peripheral blood monocyte cell.

9. Statistics: RAW 264.7 is a cell line. Thus, one would expect minimal variation in the results, making n=3 acceptable for the number of experiments run. While a presentation of the variation in 3 points as SEM is published, it is a bit misleading, especially given the very small SEM bars on Fig 3F and Fig 4.

Reply to the comment 9:

 In the present study, number of samples mean independent experiments. As our results are obtained by using cell line, variation of results is small. We have added data in Fig.3F to increase the certainty of our results (n = 5 independent experiments). 

Minor comments: 

10. English in places could be improved

• Figure legend 1 “fluorescence-labelled LPS”, Alex 594-LPS would be better.

• CD14 “is not indispensable for” is a double negative. “CD14 is necessary for” would be clearer.

Reply to the comment 10:

 Thank you for your suggestion. As you pointed out, we have revised the figure legend. The word alters “fluorescence” to “Alexa Fluor 594”. 

We have revised the sentence as follows (Page 21, Line 10-12).

In contrast to CD14, TLR4 contributes to the activation of signalling responses to LPS, although it is not indispensable for LPS uptake. 

11. Fig legend for Fig 3E; lowest concentration of 0.2 as in legend or 2.0 as in bar graph/text?

Reply to the comment 11:

 Thank you for your suggestion. We have revised figure legend (Page 26, line 16).

12. Figs 3-4 represent the novelty. Many of the controls (ie blocking of other receptors doesn’t alter LPS uptake) could be shown.

Reply to the comment 12:

 We have added supplementary figure, which shows the effect of neutralising antibody related with scavenger receptors and TLR (S3 Fig). None of neutralising antibody against SR-A, RAGE, CD36, LOX-1, TLR4, TLR5 and TLR7 has no effect on LPS uptake in RAW264.7 cells. 

 

Reviewer #3

We would like to thank reviewer #3 for the valuable comments and suggestions on our manuscript. We appreciate your quick reply to our additional inquiry. We have carefully read the comments and have made the following corrections in the revised version:

Reviewer #3: The manuscript by Kitaura et al. presents the interdependent effects of AGE3 (a particular type of advanced glycation end-product), lipopolysaccharide (LPS) and the role of the RAGE receptor for the activation of murine macrophages. The authors conclude that RAGE activation by AGE3 impairs immune response in diabetic patients.

The described experiments appear to be done carefully and support to overall conclusion of the manuscript. Weak points that should be addressed are the reliance on FACS data for AGE3 and LPS uptake, as well as for cell surface receptor expression. This could be remediated with additional experiments (see details listed below), including additional controls and alternate methodologies (e.g. qPCR / Western Blots for receptor transcription and expression and confocal microscopy for receptor localization. There are some minor correction necessary in the Material / Methods / Experimental sections (details below).

The discussion should more clearly point out the relevance of AGE 3 in diabetic patients and the limitations of translating results from cell based assays using mouse cells to human pathology and therapy.

The authors should provide additional detail regarding the following:

Materials and Methods:

1. Page 8 / line 8: What was the concentration of glycoaldehyde used to form AGE-3. How was excess glycoaldehyde removed at the end of the incubation period?

Reply to the comment 1:

We have added information of regarding concentration as follows (Page 8, line 6-12). Briefly, 50 mg/mL of BSA (Fujifilm Wako, Osaka, Japan) was incubated under sterile conditions with 0.2 M of glycolaldehyde (AGE-3) (Sigma-Aldrich) in 0.2 M phosphate buffer (pH 7.4) at 37�C for 7 days. Excess glycolaldehyde removed from AGE-BSA or BSA solution by dialysis for 2 days at 4�C. Cellulose tube 27/32 (UC27-32-100, SEKISUI CHEMICAL, Osaka, Japan) was filled with AGE-BSA or BSA solution and dialysed in 0.02 M PBS. 

2. AGE 8 line 1: What does the acronym SRL stands for?

Reply to the comment 2:

 SRL is company name and not acronym (https://www.srl-group.co.jp/english/).

3. PAGE 8: the list of antibodies used in incomplete. See page 13 for other antibodies used.

Reply to the comment 3:

 As you pointed out, all antibody used in the present study have listed in “Regents and antibodies” section (Page 9). 

 neutralising antibodies against RAGE (20 μg/ml, AF1179), LOX-1 (20 μg/ml, AF1564), SR-A/CD204 (20 μg/ml, AF1797, all R&D Systems, Minneapolis, MN, USA), CD14 (20 μg/ml, 557896, BD Biosciences, Franklin Lakes, NJ, USA), CD36 (20 μg/ml, MA5-14112, Thermo Fisher Scientific, Waltham, MA, USA), TLR4 (20 μg/ml, 117608, BioLegend, San Diego, CA, USA), TLR5 (NBP2-24787, Novus Biologicals, Centennial, CO, USA) and TLR7 (NBP2-27332, Novus Biologicals). Anti-mouse Abs against phycoerythrin (PE)-conjugated SR-A (4 ng, 130-102-328, Miltenyi Biotec, Bergisch Gladbach, Germany), PE-conjugated TLR4 (50 ng, 12-9041-80, Thermo Fisher Scientific), PE-conjugated LOX-1 (0.5 µl, FAB1564P, R&D Systems), allophycocyanin (APC)-conjugated CD14 (50 ng, 123312, BioLegend), APC-conjugated CD36 (25 ng, 102612, BioLegend), or APC-conjugated RAGE (50 ng, LS-C212626, LifeSpan BioSciences, Seattle, WA, USA) were used for FACS analysis. Isotype negative control antibodies IgG2a and IgG2b (eBioscience, San Diego, CA, USA) for FACS analysis and against anti-CD14 antibody (Rat IgG2b, 553986, BD Biosciences, Franklin Lakes, NJ, USA) and anti-RAGE antibody (Goat IgG, sc-2028, Santa Cruz Biotechnology, Dallas, TX, USA) were used.

4. PAGE 10 line 1: What was the final labeling stoichiometry of the Alexa fluorophore bound to AGE-3? BSA has only one free SH group. Is the SH group still accessible after glycation?

Reply to the comment 4:

 The final labeling stoichiometry of the Alexa Fluor fluorophore bound to AGE-3 is unclear. The reaction between AGEs and fluorescence-labelling reagent were performed according to manufacturer's protocol. In the present study, each mol of BSA or AGE-3 incubated with almost 20 mol of labeling reagent. Excess labelling reagent is removed by dialysis. AGEs are formed by the Maillard reaction followed by complex reaction including oxidation, dehydration and condensation. The Maillard reaction is generally occurred between a free amino group of protein (e.g. BSA) and carbonyl group (e.g. glycolaldehyde). The products derived from the Maillard reaction finally become AGEs via process of complex reaction. Therefore, free SH group still accessible after glycation. 

5. PAGE10 line 7/8: What does “the strength of Alexa fluor 488-BSA or AGE 3 per unit dosage were adjusted” mean? That may refer in part to Q4

Reply to the comment 5:

 Fluorescence intensity per unit dosage are different between fluorescence-labeling BSA and fluorescence-labeling AGE-3. According to previous report [17], fluorescence intensity was adjusted by addition of non-labeling BSA or AGE-3 respectively.

Reference

17. Miki Y, Dambara H, Tachibana Y, Hirano K, Konishi M, Beppu M. Macrophage recognition of toxic advanced glycosylation end products through the macrophage surface-receptor nucleolin. Biol Pharm Bull. 2014;37: 588–596. doi:10.1248/bpb.b13-00818

6. PAGE 11 line 1: Please identify and list all neutralizing antibodies in “Materials”

Reply to the comment 6:

As you pointed out, all neutralising antibody used in the present study have listed in “Regents and antibodies” section. (Please see point “Reply to the comment 3” above).

7. PAGE 12 line1: How were cells detached (mechanically or chemically) for FACS analysis.

Reply to the comment 7:

 In the present study, cells were mechanically detached by pipetting. We have revised “Materials and Methods” (Page 12, Line 3-4 and Page 13, Line 12).

8. PAGE 12 line 7 LS-C212626 and RAGE Ab appears to be a polyclonal rabbit anit0human RAGE Ab. How do the authors know that (1) the Ab recognizes mouse RAGE (2) is specific for RAGE (3) shows no cross-reactivity with other cellular proteins. This is particular concerning given that RAGE is apparently expressed at so low levels that it cannot be detected by Western blot.

Reply to the comment 8:

 You have raised an important question. In anti-RAGE antibody (LS-C212626) datasheet prepared by LSBio, reactivity to strain is human and mouse. However, we could not confirm the selectivity of this antibody, because we are not able to obtain positive control sample for FACS analysis. We have conformed RAGE expression in RAW264.7 cells by using western blot. Since LS-C212626 conjugate fluorescence substance (Allophycocyanin, APC), we used another anti-RAGE antibody (abcam, #ab3611). The #ab3611 has detected the two bands, which represent pre and post glycation type of RAGE protein, in the vicinity of 50 kDa in mouse lung tissue as positive control. The protein sample prepared from RAW264.7 cells showed weak band in the vicinity of 50 kDa, indicating that RAW264.7 cell shows much lower levels of RAGE compared to lung tissue. Niven et al reported that RAGE expression of RAW264.7 cells have confirmed at mRNA and protein levels [33]. We suggest that RAGE is a functional receptor for AGEs despite expression are low in RAW264.7 cells. 

 We have added the materials and methods and the result which indicate much lower protein expression of RAGE in RAW264.7 cells (S6 Fig). We have also revised the results and discussion.

(Results, Page 28, Line 13- Page 29, Line 5)

 We verified the protein expression of RAGE in RAW264.7 cells by using western blot. Consistent with previous study [33], the two bands, which represent pre and post glycation type of RAGE protein, in the vicinity of 50 kDa were observed in mouse lung tissue used as positive control (S6 Fig). The protein sample prepared from RAW264.7 cells showed weak band in the vicinity of 50 kDa, indicating that RAGE is a functional receptor for AGEs despite expression are low in RAW264.7 cells. Taken together, these findings suggest that AGE-3 attenuated the response to LPS in RAW264.7 cells, which was partially mediated by the activation of RAGE (Fig 5).

(Discussion, Page 32, Line 10-11)

 We detected low protein expression of RAGE in RAW264.7 cells.

Reference

33. Niven J, Hoare J, McGowan D, Devarajan G, Itohara S, Gannagé M, et al. S100B Up-Regulates Macrophage Production of IL1β and CCL22 and Influences Severity of Retinal Inflammation. Lewin AS, editor. PLoS One. 2015;10: e0132688. doi:10.1371/journal.pone.0132688

Statistical Analysis

9. The FACS data a generally scaled to a control set to 100% and values for treatments are reported % relative to controls. That means the controls have by default no error or deviation (all 100%). Doing data analysis this way can really cover up large differences between biological independent repeats. It is also not clear if any independent repeats were done. If no independent repeats were done, then scaling to 100% might be okay because then all replicates are just technical replicates. The authors should clarify this point. They should also justify the use of specific post-hoc test, under consideration that controls (defaulted to 100%) have no error.

Reply to the comment 9:

You have raised an important question. In the present study, sample size means independent experiments. In FACS experiment, we obtained data under experimental condition which MFI of untreated sample between each experiment become equal level by adjustment of laser power. There is a variation contributing to cytometer performance including laser power. We have checked cytometer performance every experiment. Therefore, there is few differences contributing to biological independent repeats and we have evaluated the relative effect of treatment.

We agree that the results including data, which defaulted to 100% and have no error, should be analyzed by appropriate statistical test. A randomized block ANOVA is superior to the one-way ANOVA with correlated and uncorrelated data [a]. However, a one-way ANOVA followed by Dunnett’s post hoc test for multiple comparisons is actually used to analysis of the result expressed as percentage of control [b]. It seems that there is no clearly established statistical analysis for data expressed percent of control.

Reference

a. Lew M. Good statistical practice in pharmacology. Problem 2. Br J Pharmacol. 2007;152: 299–303. doi:10.1038/sj.bjp.0707372

b. MD K, KL C, FX L, A D, K T, BM U, et al. Decoy exosomes provide protection against bacterial toxins. Nature. 2020;579. doi:10.1038/S41586-020-2066-6

Results:

Q10: Figure 1 (A) Explain why the untreated cells show two distinct populations, i.e. two FACS peaks.

Reply to the comment 10:

 Thank you for pointing out. We have showed the representative histogram plot by logarithmic display. But logarithmic display is not able to represent cell population showing weak fluorescence intensity and result in representation of two peaks as our figure. We have revised Fig.1A and Fig.3A. Data altered logarithmic display to biexponential display. The biexponential display is able to represent cell population showing weak fluorescence intensity and have resulted in representation of one peak. The label of x axis is logarithmic.

11. Figure 1 (A): Please label both axes of the FACS figure, in particular the fluorescence intensity. That will allow the reader to judge the magnitude of MFI changes shown in the other panels of figure 1.

Reply to the comment 11:

 We have added label of both axes in FACS figure (Fig.1A).

12. Again FACS data are shown as scaled to 100% controls. That makes it impossible to actually judge the sale of treatment related changes. This in turn makes it hard to imply that statistical significance could have biological relevance. While difference s between treatment groups appear to be of up to more than three-fold change, providing actual quantitative data would strengthen the data analysis.

Reply to the comment 12:

In FACS experiment, we obtained data under experimental condition which MFI of untreated sample between each experiment become equal level by adjustment of laser power. There is a variation contributing to cytometer performance including laser power. We have checked cytometer performance every experiment. Therefore, there is few differences contributing to biological independent repeats and we have evaluated the relative effect of treatment. Please see point “Reply to the comment 9” above.

13. Data interpretation in the written results section is mostly qualitative in nature. It strengthen the paper if the authors would discuss data on a quantitative level.

Reply to the comment 13:

 We agree with you. As you pointed out, we have revised the results to reflect the change qualitative to quantitative level in discussing data as follows.

(Page 19, Line 10-11) 

The mean florescence intensity (MFI) indicating uptake of fluorescently-labelled LPS at 1 �g/ml increased at 2 h by 268% and 4 h by 350%.

(Page 20, Line 1-3)

 In the present study, we found that LPS uptake by RAW264.7 cells was decreased by clathrin-mediated endocytosis inhibitors, sucrose at 0.45 M by 69.6% and chlorpromazine at 20 μM by 57.8% (Fig 1E).

(Page 22, Line 3-4)

 Consistent with previous reports, we confirmed that pretreatment with anti-CD14 antibody at 20 μg/ml reduced the LPS uptake by 57.1% (Fig 2B).

(Page 22, Line 13-15)

 In experimental condition which is exposed to LPS for 24 h, we found that anti-CD14 antibody at 20 μg/ml also suppressed the increased release of CXCL10 in LPS-treated RAW264.7 cells by 79.5% (Fig 2D).

(Page 25, Line 12-13)

 Moreover, AGE-3 at 200 μg/ml also reduced CXCL10 production in the presence of LPS at 1 μg/ml by 62% (Fig 3G).

14. Figure 3(A) same as Q11, figures need axes labels.

Reply to the comment 14:

We have added label of both axes in FACS figure (Fig.3A).

15. PAGE 22 line 15: Please provide a more detailed description of how the AGE3 IC50 for LPS uptake was calculated. The four values extractable from figure 3(B) are not sufficient to make such a calculation. Especially since there are not data that justify the statment “ we assumed that the effects …were maximal at 200 ug/ml”. If no additional data are available, the IC50 statement should be removed.

Reply to the comment 15:

 As you pointed put, our date is insufficient to calculate IC50. RAW264.7 cells were aggregated by exposure to AGE-3 more than 200 μg/mL, indicating that the examination of dose effect more than 200 μg/mL have difficulty. We agree with your assessment. We have delated IC50.

16. Figure 3(F): AGE 3 up to 20 ug/ml has no effect on CD14 cell surface level in the presence of LPS (Figure 3F is basically similar to figure 2A). 2 or 20 ug/ml AGE 3 also has no effect on CD14 expression (Figure 3E). IN light of this the statement on PAGE 23 line 9-11 seems poorly supported. In stead, down regulation of CXCL10 could be attributed to AGE3 alone and be independent of CD14. The following control experiment should be done: measure the effect of AGE 3 on CXCL10 (that would be an additional entry in Figure 4C)

Reply to the comment 16:

 You have raised an important question. While AGE-3 at 20 μg/mL significantly inhibited LPS uptake, no significant change in CD14 expression in the absence and presence of LPS was observed in AGE-3 at 20 μg/mL. We believe that AGE-3-induced downregulation of CD14 expression partially contributed to the reduction of LPS uptake and CXCL10 production in response to LPS. As you pointed out, our results also indicated that AGE-3 suppresses LPS uptake and CXCL10 production mediated by CD14 independent mechanisms. We appreciate your valuable comment. We have revised the text (Page 25, Line 13-Page 26, Line 1-2) as follows. 

These results suggest that the AGE-3-induced downregulation of CD14 expression partially contributed to the reduction of LPS uptake and CXCL10 production in response to LPS. Furthermore, low dose of AGE-3 inhibited LPS uptake without reduction of CD14 expression in the absence and presence of LPS, indicating that AGE-3 suppresses LPS uptake and CXCL10 production mediated by both CD14 dependent and independent mechanisms.

We have shown the effect of AGE-3 alone on the CXCL10 production in Fig.3G. AGE-3 alone has no effect on CXCL10 production. We believe that the effect of AGE-3 alone on the CXCL10 production is shown in Fig.3G but not Fig.4C, because Fig.3 have referred to the effect of AGE-3. On the other hand, Fig.4 have referred to the involvement of RAGE on the inhibitory effect of AGE-3.

17. PAGE 25 line 11/12: what is known about the target specificity / off-target effects of FPS-ZM1. Small molecule drugs are notorious for off-target effects. To confirm RAGE involvement additional experiments with monoclonal anti-RAGE antibodies should be done. In essence, the data shown in figures 4A, B and C should include an neutralizing anti-RAGE mAb, including a dose response curve (like 4A).

Reply to the comment 17:

 As you pointed out, it is possibility that FPS-ZM1 may inhibit effect of AGE via non-target effect including independent of RAGE inhibition. We have investigated the effect of anti-RAGE neutralising antibody on the AGE-3-induced reduction of LPS uptake, CD14 expression and CXCL10 production. We used polyclonal antibody but not monoclonal antibody, because we have not obtained reliable monoclonal antibody. Anti-RAGE neutralising antibody at 20 μg/mL inhibit the AGE-3-induced reduction of LPS uptake, CD14 expression and CXCL10 production. Control antibody for neutralising antibody has no effect on the CXCL10 production. In addition to results of FPS-ZM1, AGE-3 attenuated the response to LPS in RAW264.7 cells, which was partially mediated by the activation of RAGE. We could not investigate the dose-dependent effect of anti-RAGE neutralising antibody. We agree that it is important to confirm the dose-dependent effect. However, it is difficult to assess the dose-dependent effect of anti-RAGE neutralising antibody, since anti-RAGE neutralising antibody have only a small inhibitory effect.

 We have added the results of anti-RAGE antibody (Figs.4B, 4D and 4F) as follows (Page 28, Line 3-13). 

 In this study, we examined which receptor contributed to the inhibitory effect of AGE-3 on the response to LPS in RAW264.7 cells. A RAGE antagonist, FPS-ZM1 at 0.5 and 1 μM, and neutralising antibody against RAGE at 20 μg/mL prevented the AGE-3-induced downregulation of CD14 expression (Figs 4A and 4B). There was no significant effect of LPS-RS, a TLR4 antagonist, at 10 μg/mL or a neutralising antibody against SR-A (CD204) at 20 μg/mL on CD14 expression and LPS uptake in AGE-3-treated RAW264.7 cells. However, anti-SR-A antibody suppressed the uptake of AGE-3 in RAW264.7 cells (S5 Fig) indicating that the concentration of antibody used was sufficient to suppress effects mediated by AGE-3. Consistent with this, the AGE-3-induced reduction of LPS uptake and CXCL10 release were significantly inhibited by treatment with FPS-ZM1 and anti-RAGE neutralising antibody (Figs 4C-4F).

18. PAGE 28 line 9/10: What are the absolute RAGE expression levels on RAW624.7 cells? AS mentioned, the anti-RAGE Ab used for FACS seems to be inappropriate (polyclonal, anti-human RAGE). The authors should employ additional confirmatory experiments to show RAGE expression on RAW cells. Why would a Western blot not work when RAGE is there is significant quantities? If WB does not work, show IF micrographs and do an ELISA. FACS as the only method is not convincing.

Reply to the comment 18:

 You have raised an important question. Please see point “Reply to the comment 8” above.

19. Discussion: How do the results obtained from the cultured mouse cells translate to humans with diabetes? Discuss the limitations.

Reply to the comment 19:

 We appreciate your valuable comment. We agree your suggestion and have added the discussion as follows (Page 35, Line 15-16 and Page 36, Line 1-2). 

 The second limitation is that our findings are obtained by a single murine cell line. Further studies are needed to confirm the inhibitory effect of AGEs on innate immune reaction response to LPS and its molecular mechanisms using human peripheral blood monocyte cell.

20. Discussion: What is the AGE3 (not just “AGE” in general) level in diabetic vs. healthy patients? How does that compare to the 200 ug/ml used in most experiments?

Reply to the comment 20:

You have raised an important question. However, AGE-3 (glycolaldehyde derived AGE) levels in human is unclear. Measurement of AGE-3 have not been established. Glycolaldehyde complexly modify protein and several type of glycation products is formed. The serum AGEs levels in patients with diabetes is almost 80 �g/mL. Therefore, it seems that AGE-3 levels are lower than 80 �g/mL. AGEs are gradually accumulated in tissue, raising the possibility that macrophage within the local microenvironment are exposed to high AGEs levels. 

We have added the discussion about AGE-3 levels as follows (Page 31, Line14-17 and Page 32, Line 1-4).

 In the present study, we used AGE-3 at 200 μg/mL to examine the inhibitory effect of AGE-3 on the reaction response to LPS. We have no information about the AGE-3 levels in patients with diabetes. Measurement of AGE-3 have not been established. Serum AGEs levels are elevated 2~8-fold (almost 20~80 μg/mL) in patients with diabetes compared with healthy individuals [12,37]. Therefore, it seems that AGE-3 levels are lower than 80 μg/mL. AGEs are gradually accumulated in tissue, raising the possibility that macrophage within the local microenvironment are exposed to high AGEs levels in patients with diabetes.

21. Figure 5: The figure could suggest that AGE3/RAGE down-regulate CD14 on the genetic level (CD14 expression down regulation). Please clarify. To me, the figure suggests a down-regulation of CD14 on the expression level (genetic down regulation due to reduced transcription / translation). What I think the figure intends to show is the reduction of CD14 levels on the cell surface, possibly due to internalization. The two postulates are not the same. The authors should make very clear what they propose. The data presented (FACS) do not provide evidence for a genetic down-regulation of CD14.

Reply to the comment 21:

 Thank you for valuable comment. We appreciate your quick reply to our additional inquiry. We have revised the Fig.5. As you pointed out, AGE-3 suppress RAGE expression on the cell surface but not genetical levels.

---

## [Decision Letter · Decision Letter 1]

23 Nov 2020

PONE-D-20-26201R1

Advanced glycation end-products reduce lipopolysaccharide uptake by macrophages

PLOS ONE

Dear Dr. Takahashi,

Thank you for submitting your revised manuscript to PLOS ONE. After careful consideration, we feel that it does not address the concerns of the reviewers. Therefore, we invite you to submit a newly revised version of the manuscript that addresses the points raised during the review process.

We look forward to receiving your revised manuscript.

Kind regards,

David M. Ojcius

Academic Editor

PLOS ONE

Reviewers' comments:

Reviewer's Responses to Questions

**Comments to the Author**

1. If the authors have adequately addressed your comments raised in a previous round of review and you feel that this manuscript is now acceptable for publication, you may indicate that here to bypass the “Comments to the Author” section, enter your conflict of interest statement in the “Confidential to Editor” section, and submit your "Accept" recommendation.

Reviewer #1: All comments have been addressed

Reviewer #2: (No Response)

Reviewer #3: (No Response)

2. Is the manuscript technically sound, and do the data support the conclusions?

Reviewer #1: Yes

Reviewer #2: Partly

Reviewer #3: No

3. Has the statistical analysis been performed appropriately and rigorously? 

Reviewer #1: Yes

Reviewer #2: Yes

Reviewer #3: No

4. Have the authors made all data underlying the findings in their manuscript fully available?

Reviewer #1: Yes

Reviewer #2: Yes

Reviewer #3: No

5. Is the manuscript presented in an intelligible fashion and written in standard English?

Reviewer #1: Yes

Reviewer #2: Yes

Reviewer #3: Yes

6. Review Comments to the Author

Reviewer #1: The authors of manuscript ID PONE-D-20-26201R1, "Advanced glycation end-products reduce lipopolysaccharide uptake by macrophages" have addressed my prior comments and questions.

Reviewer #2: The authors addressed many of the reviewer’s comments although a better discussion of the findings and thoughts on mechanism by which AGE/RAGE impact CD14 and CXCL10 would considerably strengthen the manuscript.

Specifically, the validation of the anti-RAGE neutralizing Ab. That the Ab recognizes two large RAGe bands in a positive control is good. However, there are no specific bands corresponding to RAGE in the RAW cells. With this Ab, there are many bands and the most prominent bands are not at 50 kDa. That, coupled with a lack of documentation that the Ab actually binds to anything on the cells (can the receptor be detected by flow?), leaves only the FPS-ZM1 data to implicate RAGE. As FPS-ZM1 is a pharmacological approach, the data is suggestive (could it be blocking something other receptor?, binding and stabilizing CD14 at the cell surface?) but not definitive.

Missing from the discussion/model is any suggestion of mechanism. The data in Figure 2 are intriguing. CD14 levels return to normal by 4 h but it takes 24 hr to get significant release of CXCL10. Why is LPS necessary for >16 h when CD14 is restored much earlier? LPS must be doing something more than just reducing CD14 levels. Are there any suggestions on what that might be? How do the authors envision that AGE-3/RAGE engagement reduces CD14 expression and CXCL10 production? And how might AGE/RAGE be impacting the normal LPS signaling for CXCL10?

Minor comments

No need to define MFI in every figure. In methods and Fig 1 is fine. Same with "n= means the number of exp."

Page 22 line 11-12: “not indispensable for LPS uptake”. This means that TLR4 is REQUIRED for endocytosis. I think the authors mean “is not required for LPS uptake” Please replace the phrase “not indispensable” with a clearer description. That is, is TLR4 required for endocytosis? If so, say that. If it is not, then state that clearly that it is not necessary.

Page 23 line 7 confirmed, not conformed

Page 23, line 8, but, not bot

Page 26-27 lines 16 and 1-2 “Furthermore, low dose of AGE-3 inhibited LPS uptake without reduction of CD14 expression in the absence and presence of LPS, indicating that AGE-3 suppresses LPS uptake and CXCL10 production mediated by both CD14 dependent and independent mechanisms.” From Fig 3(B), 20 µg/mL AGE3 significantly inhibits LPS uptake AND there is a trend (not statistically significant but this is only an n=3, Fig 3E) towards reduced CD14. In Fig 3F, the MFIs are so low (and flow is read on a log scale), the differences seen (or not seen) by flow could be a limitation of the sensitivity of the flow machine. In this reviewer’s opinion, the author’s conclusion is not supported by the data and should be removed from the final version. It is noted that the n on Fig 3F is 5 vs 3 for most of the other data. Was this done more times to see if the results were significant?

Page 29, lines 9-11: The fact that anti-SR-A Ab suppresses AGE-3 uptake does not confirm that the same concentration of a different (anti-RAGE) Ab would have the same effect. Granted they are both polyclonal goat IgGs but they are not the same Ab and getting blocking with one cannot be generalized to all Abs. This is an overinterpretation of the data.

Page 29: “We verified the protein expression of RAGE in RAW264.7 cells by using western blot. Consistent with previous study [33], the two bands, which represent pre and post glycation type of RAGE protein, in the vicinity of 50 kDa were observed in mouse lung tissue used as positive control (S6 Fig). The protein sample prepared from RAW264.7 cells showed weak band in the vicinity of 50 kDa, indicating that RAGE is a functional receptor for AGEs despite expression are low in RAW264.7 cells.” The very light band at ~50K in RAW cells does not convincingly show that RAGE is expressed in this cell line. The gel seems overexposed as indicated by the very dark RAGE bands in the positive control. A lower exposure would show no bands in the 50 KDa range. Even so, assuming this clone of RAW cells is the same as that reported in ref 33, a PCR showing message expression would be more convincing than the Western shown. Additionally, the fact that there may be RAGE expression in these cells (questionable as this is not a clean gel and there are many bands of equal or higher intensity to the putative RAGE band), the presence of protein does not demonstrate that RAGE is a FUNCTIONAL receptor.

Page 36: What do the authors mean that “distribution of cell population was confused”?

Finally, the English still needs to corrected. Some of the edits to "correct" English are wrong. Please have native English speaker (or editor) correct so that verb and subject match.

Reviewer #3: I have very carefully read the authors' responses to the reviewers and the revised manuscript. I do not feel that manuscript has substantially improved in scientific rigor. The data still have weaknesses which make the interpretation of the experimental results ambiguous. The authors acknowledge this fact in their responses repeatedly (the reviewers rise a valid point), but they do not address these shortcomings experimentally, but rather linguistically. At this stage I do not support publication in PLOS ONE

7. PLOS authors have the option to publish the peer review history of their article (what does this mean?). If published, this will include your full peer review and any attached files.

Reviewer #1: No

Reviewer #2: No

Reviewer #3: No

---

## [Author Response · Author response to Decision Letter 1]

24 Dec 2020

Response to reviewers:

Reviewer #2

We would like to express our deepest gratitude reviewer #2 for the valuable comments and suggestions on our manuscript. It was really helpful for us and all suggestions allowed us to improve the quality of our manuscript. 

1. The authors addressed many of the reviewer’s comments although a better discussion of the findings and thoughts on mechanism by which AGE/RAGE impact CD14 and CXCL10 would considerably strengthen the manuscript. Specifically, the validation of the anti-RAGE neutralizing Ab. That the Ab recognizes two large RAGe bands in a positive control is good. However, there are no specific bands corresponding to RAGE in the RAW cells. With this Ab, there are many bands and the most prominent bands are not at 50 kDa. That, coupled with a lack of documentation that the Ab actually binds to anything on the cells (can the receptor be detected by flow?), leaves only the FPS-ZM1 data to implicate RAGE. As FPS-ZM1 is a pharmacological approach, the data is suggestive (could it be blocking something other receptor?, binding and stabilizing CD14 at the cell surface?) but not definitive.

Reply to the comment 1:

 Anti-RAGE antibody is able to detect RAGE protein, because the results from western blot analysis showed clearly positive bands in tissue lysate of mouse lung. However, as you pointed out, non-related bands with RAGE were observed in cell lysate of RAW264.7 cells. It is possible that anti-RAGE antibody exhibits inhibitory effect on AGEs though RAGE-non-dependent target. FPS-ZM1 and anti-RAGE antibody partially inhibit AGE-induced downregulation of CXCL10 release from LPS-treated cells, indicating that other target molecules are involved in AGEs effect. High concentration of FPS-ZM1 at 10 μM has inhibitory effect on androgen receptor, benzodiazepine receptor, kappa opioid receptor and melatonin receptor. However, none of these receptors bind to AGEs. In addition, our results showed that FPS-ZM1 alone did not affect LPS-induced CXCL10 release (Fig. 4E). Although our results remain some limitation, we considered that FPS-ZM1 and anti-RAGE antibody inhibit AGEs though blockage of RAGE in the main. We have revised the discussion as follows. 

(Page 32, Line 14-16 – Page 33, Line 1)

Although we have not clearly detected RAGE protein in RAW264.7 cells compared to lung tissue as positive control, AGE-3-induced the suppression of response to LPS was attenuated by FPS-ZM1 and neutralising antibody against RAGE. The RAGE expression in RAW264.7 cells may therefore be quite low.

(Page 37, Line14-16 – Page 38, Line 1-7)

The second limitation is that the evidence for involvement of RAGE on AGE-induced suppression of response to LPS have not adequately established. In western blot analysis for protein expression of RAGE, indistinct bands were observed in lysate of RAW264.7 cells in contrast to lung tissue. In the present study, we did not confirm the involvement of RAGE using knockout or knockdown experiments. High concentration of FPS-ZM1 at 10 μM has inhibitory effect on androgen receptor, benzodiazepine receptor, kappa opioid receptor and melatonin receptor [55]. However, none of these receptors bind to AGEs. In addition, our results showed that FPS-ZM1 alone did not affect LPS-induced CXCL10 release. Although our results remain limitation, we considered that FPS-ZM1 and anti-RAGE antibody inhibit AGEs though blockage of RAGE in the main.

Reference

[55] Drake LR, Brooks AF, Stauff J, Sherman PS, Arteaga J, Koeppe RA, et al. Strategies for PET imaging of the receptor for advanced glycation endproducts (RAGE). J Pharm Anal. 2020;10: 452–465. doi:10.1016/J.JPHA.2020.07.009

2. Missing from the discussion/model is any suggestion of mechanism. The data in Figure 2 are intriguing. CD14 levels return to normal by 4 h but it takes 24 hr to get significant release of CXCL10. Why is LPS necessary for >16 h when CD14 is restored much earlier? LPS must be doing something more than just reducing CD14 levels. Are there any suggestions on what that might be? How do the authors envision that AGE-3/RAGE engagement reduces CD14 expression and CXCL10 production? And how might AGE/RAGE be impacting the normal LPS signaling for CXCL10?

Reply to the comment 2:

 We have revised the discussion as follows (Page 34, Line12-16 – Page 35, Line 1-7).

In general, LPS activates TRIF-dependent signalling and facilitates the transcription of CXCL10 mediated by the phosphorylation of IRF3. Previous report has demonstrated that upregulation of CXCL10 mRNA by LPS began to increase and reached maximal level at 4 h in peritoneal macrophage prepared from mouse [27]. And then CXCL10 protein are produced by translation and released into culture medium. Therefore, we suggest that the time lag is occurred between CD14-initiated LPS uptake and CXCL10 production. Additionally, LPS activate release other proinflammatory cytokines such as TNF-α in macrophage. The TNF-α release from macrophage cells occur within a short period (<4hr) after exposure to LPS [46,47]. TNF-α also increases CXCL10 production in macrophage [48]. It raises possibility that TNF-α released from RAW264.7 cells treated with LPS synergically induce CXCL10 release after long-term exposure to LPS.

(Page 35, Line9-16 – Page 36, Line 1-8)

In the present study, our results suggest that AGEs-induce downregulation of CD14 expression is partially involved in reduction of CXCL10 release in RAW264.7 cells treated with LPS. LPS decreases CD14 expression on cell surface, since LPS binding to CD14 is endocytosed into cell. Although uptake of AGEs is occurred in RAW264.7 cells, anti-CD14 antibody did not affect AGEs uptake, indicating that CD14 independent mechanisms contribute to AGEs uptake. Our previous report has demonstrated that SR-A is associated with AGEs uptake in RAW264.7 cells [15]. It has been reported that stimulation of monocyte by activating agents such as LPS and IFN-γ results in downregulation of CD14 expression on cell surface mediated by shedding of CD14 [49]. CD14 is anchored to the cell membrane by glycosylphosphatidylinositol-anchor [50]. Protease including matrix metalloproteinases (MMPs) contribute to the proteolytic shedding of CD14 [51]. AGE/RAGE signaling increases expression and activity of MMP-9 in macrophage [52]. Therefore, one possibility is that AGE/RAGE decreases CD14 expression on cell surface through excessive shedding of CD14, resulting in decreased CXCL10 production after exposure to LPS.

Reference

[46] Thorley AJ, Ford PA, Giembycz MA, Goldstraw P, Young A, Tetley TD. Differential Regulation of Cytokine Release and Leukocyte Migration by Lipopolysaccharide-Stimulated Primary Human Lung Alveolar Type II Epithelial Cells and Macrophages. J Immunol. 2007;178: 463–473. doi:10.4049/JIMMUNOL.178.1.463

[47] TA H, DM T, SK D, AK C, JW P, A B. Post-transcriptional effects of extracellular pH on tumour necrosis factor-alpha production in RAW 246.7 and J774 A.1 cells. Clin Sci (Lond). 2001;100: 259–266. Available: https://pubmed.ncbi.nlm.nih.gov/11222111/

[48] Qi X-F, Kim D-H, Yoon Y-S, Jin D, Huang X-Z, Li J-H, et al. Essential involvement of cross-talk between IFN-γ and TNF-α in CXCL10 production in human THP-1 monocytes. J Cell Physiol. 2009;220: 690–697. doi:10.1002/jcp.21815

[49] Bazil V, Strominger JL. Shedding as a mechanism of down-modulation of CD14 on stimulated human monocytes. J Immunol. 1991;147: 1567–1574. Available: https://www.jimmunol.org/content/147/5/1567

[50] Haziot A, Chen S, Ferrero E, Low MG, Silber R, Goyert SM. The monocyte differentiation antigen, CD14, is anchored to the cell membrane by a phosphatidylinositol linkage. J Immunol. 1988;141: 547–552. Available: https://www.jimmunol.org/content/141/2/547.short

[51] Senft AP, Korfhagen TR, Whitsett JA, Shapiro SD, LeVine AM. Surfactant Protein-D Regulates Soluble CD14 through Matrix Metalloproteinase-12. J Immunol. 2005;174: 4953–4959. doi:10.4049/JIMMUNOL.174.8.4953

[52] Zhang F, Banker G, Liu X, Suwanabol PA, Lengfeld J, Yamanouchi D, et al. The Novel Function of Advanced Glycation End Products in Regulation of MMP-9 Production. J Surg Res. 2011;171: 871–876. doi:10.1016/J.JSS.2010.04.027

Minor comments

3. No need to define MFI in every figure. In methods and Fig 1 is fine. Same with "n= means the number of exp."

Reply to the comment 3:

 As you pointed out, MFI and number of samples define in methods and Fig.1. 

4. Page 22 line 11-12: “not indispensable for LPS uptake”. This means that TLR4 is REQUIRED for endocytosis. I think the authors mean “is not required for LPS uptake” Please replace the phrase “not indispensable” with a clearer description. That is, is TLR4 required for endocytosis? If so, say that. If it is not, then state that clearly that it is not necessary.

Reply to the comment 4:

 Thank you for valuable comment. TLR4 is not required for LPS uptake. We have revised the phrase (Page 22, Line 9-11). 

CD14 is an essential protein for LPS recognition and uptake. In contrast to CD14, TLR4 contributes to the activation of signalling responses to LPS, although it is not required for LPS uptake [24].

5. Page 23 line 7 confirmed, not conformed

Reply to the comment 5:

 As you pointed out, we have revised the phrase. 

6. Page 23, line 8, but, not bot

Reply to the comment 6:

As you pointed out, we have revised the phrase.

7. Page 26-27 lines 16 and 1-2 “Furthermore, low dose of AGE-3 inhibited LPS uptake without reduction of CD14 expression in the absence and presence of LPS, indicating that AGE-3 suppresses LPS uptake and CXCL10 production mediated by both CD14 dependent and independent mechanisms.” From Fig 3(B), 20 µg/mL AGE3 significantly inhibits LPS uptake AND there is a trend (not statistically significant but this is only an n=3, Fig 3E) towards reduced CD14. In Fig 3F, the MFIs are so low (and flow is read on a log scale), the differences seen (or not seen) by flow could be a limitation of the sensitivity of the flow machine. In this reviewer’s opinion, the author’s conclusion is not supported by the data and should be removed from the final version. It is noted that the n on Fig 3F is 5 vs 3 for most of the other data. Was this done more times to see if the results were significant?

Reply to the comment 7:

 Unlike inhibitory effect of anti-CD14 antibody, AGE-3 slightly decreases LPS uptake and CD14 expression in the presence of LPS. Especially, the effect of low-dose AGE-3 on CD14 expression is controversial. As you pointed out, our results relating to CD14 expression are only obtained by flow cytometry. Other experimental techniques such as immunocytochemistry or western blot for membrane protein are seem necessary to evaluate the effect of low-dose AGE-3 on CD14 expression. We have deleted the conclusion relating to the effect of low-dose AGE-3.

8. Page 29, lines 9-11: The fact that anti-SR-A Ab suppresses AGE-3 uptake does not confirm that the same concentration of a different (anti-RAGE) Ab would have the same effect. Granted they are both polyclonal goat IgGs but they are not the same Ab and getting blocking with one cannot be generalized to all Abs. This is an overinterpretation of the data.

Reply to the comment 8:

In Page 29, lines 9-11, we would like to have described that SR-A do not contribute to the AGE-3-induced suppression of LPS uptake and CD14 expression. Figs 4A and 4C show that anti-SR-A antibody at 20 µg/mL failed to inhibit AGE-3-induced suppression of LPS uptake and CD14 expression. On the other hand, anti-SR-A antibody at 20 µg/mL suppressed AGE-3 uptake. Therefore, our findings suggest that dose of anti-SR-A antibody at 20 µg/mL are sufficient to suppress effects mediated by AGE-3. In our previous study, we have demonstrated that anti-RAGE antibody at 20 µg/mL failed to inhibit AGE-3 uptake in RAW264.7 cells. We have revised the sentence as follows (Page 28, Line 16 – Page 29, Line 1-3). 

However, anti-SR-A antibody suppressed the uptake of AGE-3 in RAW264.7 cells (S5 Fig) indicating that the concentration of anti-SR-A antibody sufficient to suppress effects mediated by AGE-3. Therefore, SR-A do not contribute to the AGE-3-induced suppression of LPS uptake and CD14 expression. 

9. Page 29: “We verified the protein expression of RAGE in RAW264.7 cells by using western blot. Consistent with previous study [33], the two bands, which represent pre and post glycation type of RAGE protein, in the vicinity of 50 kDa were observed in mouse lung tissue used as positive control (S6 Fig). The protein sample prepared from RAW264.7 cells showed weak band in the vicinity of 50 kDa, indicating that RAGE is a functional receptor for AGEs despite expression are low in RAW264.7 cells.” The very light band at ~50K in RAW cells does not convincingly show that RAGE is expressed in this cell line. The gel seems overexposed as indicated by the very dark RAGE bands in the positive control. A lower exposure would show no bands in the 50 KDa range. Even so, assuming this clone of RAW cells is the same as that reported in ref 33, a PCR showing message expression would be more convincing than the Western shown. Additionally, the fact that there may be RAGE expression in these cells (questionable as this is not a clean gel and there are many bands of equal or higher intensity to the putative RAGE band), the presence of protein does not demonstrate that RAGE is a FUNCTIONAL receptor.

Reply to the comment 9:

 RAGE is abundantly expressed in lung tissue compared to other tissue. From database of gene expression profile (BioGPS: http://biogps.org), RAGE expression in lung tissue is almost two thousand-fold higher than other tissue or cells including RAW264.7 cells. Our results obtained from western blot show RAGE bands has little detected in RAW264.7 cells under lower exposure condition. Although source of RAW264.7 cells using ref 33 is unclear, protein expression is important for evaluate receptor function compared to mRNA expression. In the present study, we found that both FPS-ZM1 and anti-RAGE antibody attenuate AGE-3-induced suppression of LPS uptake, CD14 expression and CXCL10 release form RAW264.7 cells treated with LPS. However, we did not confirm the involvement of RAGE using knockout or knockdown experiments. The limitation of the present study is lack of evidence for involvement of RAGE on the AGE-induced suppression of LPS response inRAW264.7 cells. Although both FPS-ZM1 and anti-RAGE antibody attenuate the effect of AGE-3, protein expression of RAGE in RAW264.7 cells is unclear. We did not confirm the involvement of RAGE using knockout or knockdown experiments. We have added the discussion ((Page 38, Line16 – Page 39, Line 1-9). Please see point “Reply to the comment 1” above.

10. Page 36: What do the authors mean that “distribution of cell population was confused”?

Reply to the comment 10:

 In flow cytometric analysis using cell line, distribution of cell population based on cell size and intracellular complexity converge one population, because cells are homogeneous and has uniform size. LPS treatment results in confusion of cell distribution. It is possibility that the LPS treatment or the experimental techniques, which prepare cell sample for flow cytometric analysis, potentially damage cell. We considered that confusion of cell distribution induced by LPS treatment mean cell damage. We have added the text (Page 37, Line 6-7).

11. Finally, the English still needs to corrected. Some of the edits to "correct" English are wrong. Please have native English speaker (or editor) correct so that verb and subject match.

Reply to the comment 11:

 We have carefully corrected our manuscript.

---

## [Decision Letter · Decision Letter 2]

12 Jan 2021

Advanced glycation end-products reduce lipopolysaccharide uptake by macrophages

PONE-D-20-26201R2

Dear Dr. Takahashi,

We’re pleased to inform you that your manuscript has been judged scientifically suitable for publication and will be formally accepted for publication once it meets all outstanding technical requirements.

Kind regards,

David M. Ojcius

Academic Editor

PLOS ONE

Additional Editor Comments (optional):

Reviewers' comments:

Reviewer's Responses to Questions

**Comments to the Author**

1. If the authors have adequately addressed your comments raised in a previous round of review and you feel that this manuscript is now acceptable for publication, you may indicate that here to bypass the “Comments to the Author” section, enter your conflict of interest statement in the “Confidential to Editor” section, and submit your "Accept" recommendation.

Reviewer #1: All comments have been addressed

Reviewer #3: All comments have been addressed

2. Is the manuscript technically sound, and do the data support the conclusions?

Reviewer #1: Partly

Reviewer #3: Yes

3. Has the statistical analysis been performed appropriately and rigorously? 

Reviewer #1: Yes

Reviewer #3: Yes

4. Have the authors made all data underlying the findings in their manuscript fully available?

Reviewer #1: No

Reviewer #3: Yes

5. Is the manuscript presented in an intelligible fashion and written in standard English?

Reviewer #1: Yes

Reviewer #3: Yes

6. Review Comments to the Author

Reviewer #1: The authors of manuscript ID PONE-D-20-26201R2, "Advanced glycation end-products reduce lipopolysaccharide uptake by macrophages" have addressed my prior comments.

Reviewer #3: I think the authors have done a decent job addressing and responding to the comments. The manuscript has substantially improved and is acceptable for publications.

7. PLOS authors have the option to publish the peer review history of their article (what does this mean?). If published, this will include your full peer review and any attached files.

Reviewer #1: No

Reviewer #3: No

---

## [Editor Report · Acceptance letter]

14 Jan 2021

PONE-D-20-26201R2 

Advanced glycation end-products reduce lipopolysaccharide uptake by macrophages 

Dear Dr. Takahashi:

I'm pleased to inform you that your manuscript has been deemed suitable for publication in PLOS ONE. Congratulations! Your manuscript is now with our production department. 

Kind regards, 

on behalf of

Dr. David M. Ojcius 

Academic Editor

PLOS ONE